# Natural Tr1-like cells do not confer long-term tolerogenic memory

Koshika Yadava[1,2†*], Carlos Obed Medina[1†], Heather Ishak[1], Irina Gurevich[1], Hedwich Kuipers[1,3], Elya Ali Shamskhou[1], Ievgen O Koliesnik[1], James J Moon[4,5], Casey Weaver[6], Kari Christine Nadeau[7], Paul L Bollyky[1]

[1]Division of Infectious Diseases and Geographic Medicine, Department of Medicine, Beckman Center, Stanford University School of Medicine, Stanford, United States; [2]Radcliffe Department of Medicine, Weatherall Institute of Molecular Medicine, University of Oxford, Oxford, United Kingdom; [3]Department of Clinical Neurosciences, University of Calgary, Calgary, Canada; [4]Center for Immunology and Inflammatory Diseases, Massachusetts General Hospital, Harvard Medical School, Charlestown, United States; [5]Division of Pulmonary and Critical Care Medicine, Massachusetts General Hospital, Harvard Medical School, Charlestown, United States; [6]Bevill Biomedical Research Building, The University of Alabama at Birmingham, Birmingham, United States; [7]Sean N Parker Center for Allergy & Asthma Research, Stanford University, Mountain View, United States

*For correspondence:
koshika.yadava@gmail.com

†These authors contributed equally to this work

Competing interests: The authors declare that no competing interests exist.

**Abstract** IL-10-producing Tr1 cells promote tolerance but their contributions to tolerogenic memory are unclear. Using 10BiT mice that carry a Foxp3-eGFP reporter and stably express CD90.1 following IL-10 production, we characterized the spatiotemporal dynamics of Tr1 cells in a house dust mite model of allergic airway inflammation. CD90.1+Foxp3-IL-10+ Tr1 cells arise from memory cells and rejoin the tissue-resident memory T-cell pool after cessation of IL-10 production. Persistent antigenic stimulation is necessary to sustain IL-10 production and *Irf1* and *Batf* expression distinguishes CD90.1+Foxp3-IL-10+ Tr1 cells from CD90.1+Foxp3-IL-10- 'former' Tr1. Depletion of Tr1-like cells after primary sensitization exacerbates allergic airway inflammation. However, neither transfer nor depletion of former Tr1 cells influences either Tr1 numbers or the inflammatory response during subsequent allergen memory re-challenge weeks later. Together these data suggest that naturally-arising Tr1 cells do not necessarily give rise to more Tr1 upon allergen re-challenge or contribute to tolerogenic memory. This phenotypic instability may limit efforts to re-establish tolerance by expanding Tr1 in vivo.
DOI: https://doi.org/10.7554/eLife.44821.001

## Introduction

Allergic asthma is a common childhood illness and can be triggered by exposure to aeroallergens such as house dust mite (HDM). Hallmark features of the disease include aberrant T helper 2 (Th2) type responses, airway hyperreactivity, eosinophilic inflammation, increased IgE and mucus hypersecretion (*Finkelman et al., 2010*).

The balance between pro-inflammatory CD4+ Th2 cells and regulatory T cells, including CD4 +Foxp3+ regulatory T cells (Treg) and CD4+Foxp3-, interleukin-10 (IL-10)-producing Type one regulatory T cells (Tr1) cells, is a significant determinant in the development of allergic disease (*Robinson, 2009*). Allergen-specific immunotherapy may re-establish tolerance in part by expanding these regulatory T cell populations (*Akdis and Akdis, 2014*; *Meiler et al., 2008*). There is therefore great

interest in developing stable, antigen specific regulatory T cells for treatment of asthma and allergies (*Bassirpour and Zoratti, 2014*).

The anti-inflammatory cytokine interleukin 10 (IL-10) is critical for immune tolerance to airway allergens (*Hawrylowicz, 2005*). The suppressive capabilities of Tr1 cells are primarily attributed to IL-10 while Foxp3+ T cells exert regulatory function in several ways, including via IL-10 production, (*Lloyd and Hawrylowicz, 2009*; *Kearley et al., 2012*; *Wilson et al., 1992*; *Kearley et al., 2005*; *Pellerin et al., 2014*). IL-10 can limit both Th2 differentiation and survival (*Coomes et al., 2017*) and endogenous T cell-derived IL-10 limits the development of pathogenic Th2 responses during allergen sensitization (*Wilson et al., 2007*; *Tournoy et al., 2000*). Indeed, the development of Tr1-like cells is critical to the success of allergen-specific immunotherapy (*Akdis and Akdis, 2014*; *Meiler et al., 2008*). There is therefore great interest in the factors involved in Tr1 development (*Levings et al., 2005*; *Brockmann et al., 2017*; *Gagliani et al., 2013*; *Wu et al., 2011*) and considerable progress has been made toward engineering Tr1-like cells in vitro for potential therapeutic applications (*Gregori and Roncarolo, 2018*).

However, much about naturally-arising Tr1 cells and endogenous IL-10 production in allergic airway disease remains unknown (*White and Wraith, 2016*). In particular, the contribution of antigen-specific Tr1 cells to tolerogenic memory is unclear. Given the persistence of allergen-specific immunity (*Hondowicz et al., 2016*) this information may prove vital for the development of successful Tr1-directed immunotherapies.

Here, we characterize the spatial and temporal dynamics of endogenous IL-10-producing T cells in a mouse model of HDM-induced allergic airway inflammation. For this purpose, we have used a mouse strain carrying a stable IL-10 reporter, the 10BiT mouse (*Maynard et al., 2007*). This strain carries multiple copies of a bacterial artificial chromosome (BAC) transgene containing a CD90.1 construct under the control of the IL-10 promoter such that cells that previously made IL-10 express CD90.1 which persists for some time after cessation of IL-10 production. This strain was previously crossed against the Foxp3-eGFP mouse strain to facilitate discrimination between Foxp3- Tr1 from Foxp3+ Treg (*Maynard et al., 2007*).

Using these animals, we have analyzed the population frequencies and cytokine production profiles of Tr1-like cells in multiple tissues at various stages of allergic airway inflammation including sensitization, challenge, resolution, and memory. To determine the functional contribution of Tr1-like memory T-cell populations we perform loss and gain of function studies by deleting these cells or adoptively transferring them before re-challenge with the allergen. Together, these studies elucidate the cellular source of IL-10 in allergic inflammation, their functional stability, and their contribution to tolerogenic memory in the lung.

## Results

### IL-10-producing T cells accumulate at site of allergen sensitization

To interrogate the patterns of endogenous IL-10 production in allergic airway inflammation we used a house dust mite (HDM) induced murine model (*Figure 1A*). In this model, animals are sensitized to crude house dust mite (HDM) protein intranasally (i.n.) over 2 weeks while control animals are given phosphate buffered saline (PBS). Analysis of the inflammatory response is carried out 4 days after completion of the challenge series. This model recapitulates key features of allergic responses associated with type two inflammation, with HDM- sensitized animals showing an increase in total number of cells (*Figure 1B*) and total number of eosinophils (*Figure 1C*) in the broncho-alveolar lavage (BAL) as compared to controls. Also, the lungs of HDM-sensitized animals exhibited an increase in peribronchiolar and perivascular inflammation (*Figure 1D*). The levels of total IgE were also elevated in the BAL supernatants obtained from HDM-sensitized animals (*Figure 1E*).

Next, using the 10BiT/Foxp3eGFP strain (*Tg(Il10-Thy1)1Weav* crossed to *Foxp3$^{tm2Tch}$*) (*Maynard et al., 2007*), we characterized the endogenous IL-10 response in this model. This strain carries dual reporters for IL-10 and Foxp3. A bacterial artificial chromosome (BAC) containing CD90.1 under the control of an IL-10 promoter enables the detection of cells that have produced IL-10 via the expression of cell surface CD90.1 while a GFP knocked into the Foxp3 locus tracks endogenous Foxp3 expression (*Maynard et al., 2007*). Using this model, we found that the frequency of CD90.1+ Foxp3-CD4+ Tr1-like cells was increased in the HDM-sensitized animals in the BAL and the

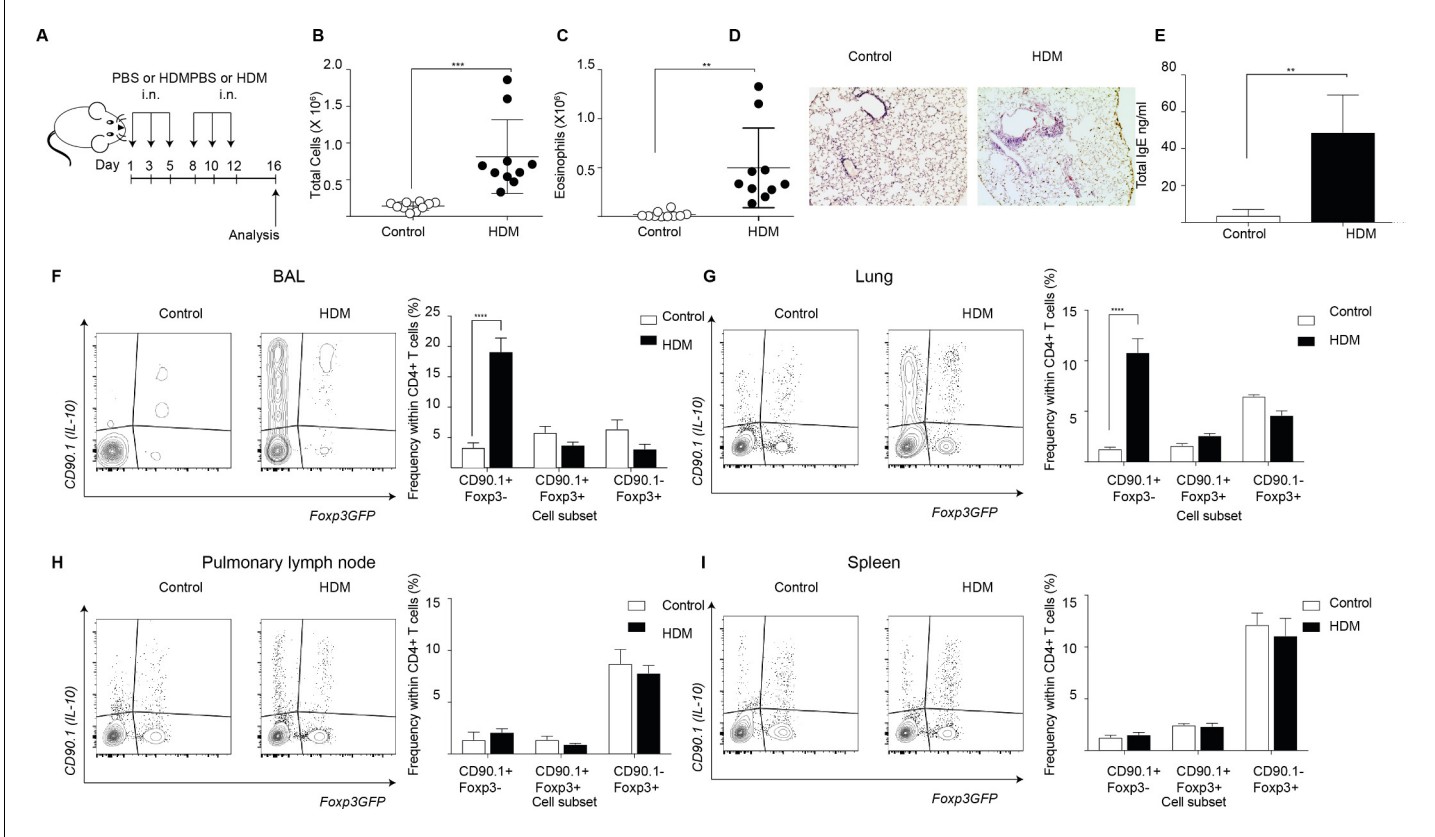

**Figure 1.** IL-10-producing T cells accumulate at site of allergen sensitization. C57Bl/6 mice were administered either PBS as control or crude HDM containing 20µg total protein in 50µl intranasally (i.n.) Six times over 2 weeks as shown. Terminal analysis was performed 4 days after last challenge. (B) The total number of cells and (C) total number of eosinophils in the BAL were determined. Data are pooled from two experiments. Each symbol represents a single animal. Error bars represent standard error of mean. n=10 for control group, n=10 for HDM group. (D) Representative haematoxylin and eosin stained lung sections showing perivascular and peribronchiolar inflammation in PBS control and HDM-sensitized animals. (E) The level of IgE in BAL supernatant was determined by ELISA. n=5 for control group, n=5 for HDM group. Error bars represent standard deviation of mean. Statistical significance was determined using an unpaired two tailed students t-test. Representative flow cytometry plots showing different T cell subsets as identified by surface expression of CD90.1 and Foxp3eGFP in the (F) BAL, (G) lungs, (H) pulmonary lymph nodes, and (I) spleens of control or HDM-treated mice 4 days after final challenge. The frequency of IL-10-producing Foxp3- (CD90.1+Foxp3-), IL-10-producing Foxp3+ (CD90.1+Foxp3+) and Foxp3+ cells which do not produce IL-10 (CD90.1-Foxp3+) within all CD4+ T cells in the four sites are plotted. Data are pooled from two experiments. Error bars represent standard error of mean. n=10 for control group, n=10 for HDM group. Data are representative of 4 independent experiments. Statistical significance was determined using 2-way ANOVA (post hoc test: Sidaks). *P≤0.05, **P≤0.01, ***P≤0.001 ****P≤0.0001. PBS=Phosphate buffered saline, HDM=house dust mite, i.n=intranasally, BAL=bronchoalveolar lavage.

DOI: https://doi.org/10.7554/eLife.44821.002
The following source data is available for figure 1:

**Source data 1.** IL-10-producing T cells accumulate at site of allergen sensitization.
DOI: https://doi.org/10.7554/eLife.44821.003

lungs (*Figure 1F,G*), the sites of allergen challenge, but not in the pulmonary lymph node (*Figure 1H*) or the spleen (*Figure 1I*). Thus, CD90.1+Foxp3- Tr1-like cells specifically accumulated at the site of allergen challenge.

## Lung Tr1-like cells are located in the lung parenchyma

We used intravenous labeling to further discriminate between lung-resident versus circulating cells (*Anderson et al., 2014*). For staining non-tissue-resident circulating cells in the lung, we injected CD45 antibody retro-orbitally 2 min before mice were euthanized. We found that the majority of CD90.1+Foxp3-CD4+ T cells were parenchymal (*Figure 2A–D*) as were the CD90.1+Foxp3+CD4+ T cells (*Figure 2E*). In contrast more than half of CD90.1-Foxp3+CD4+ T cells and about half of the

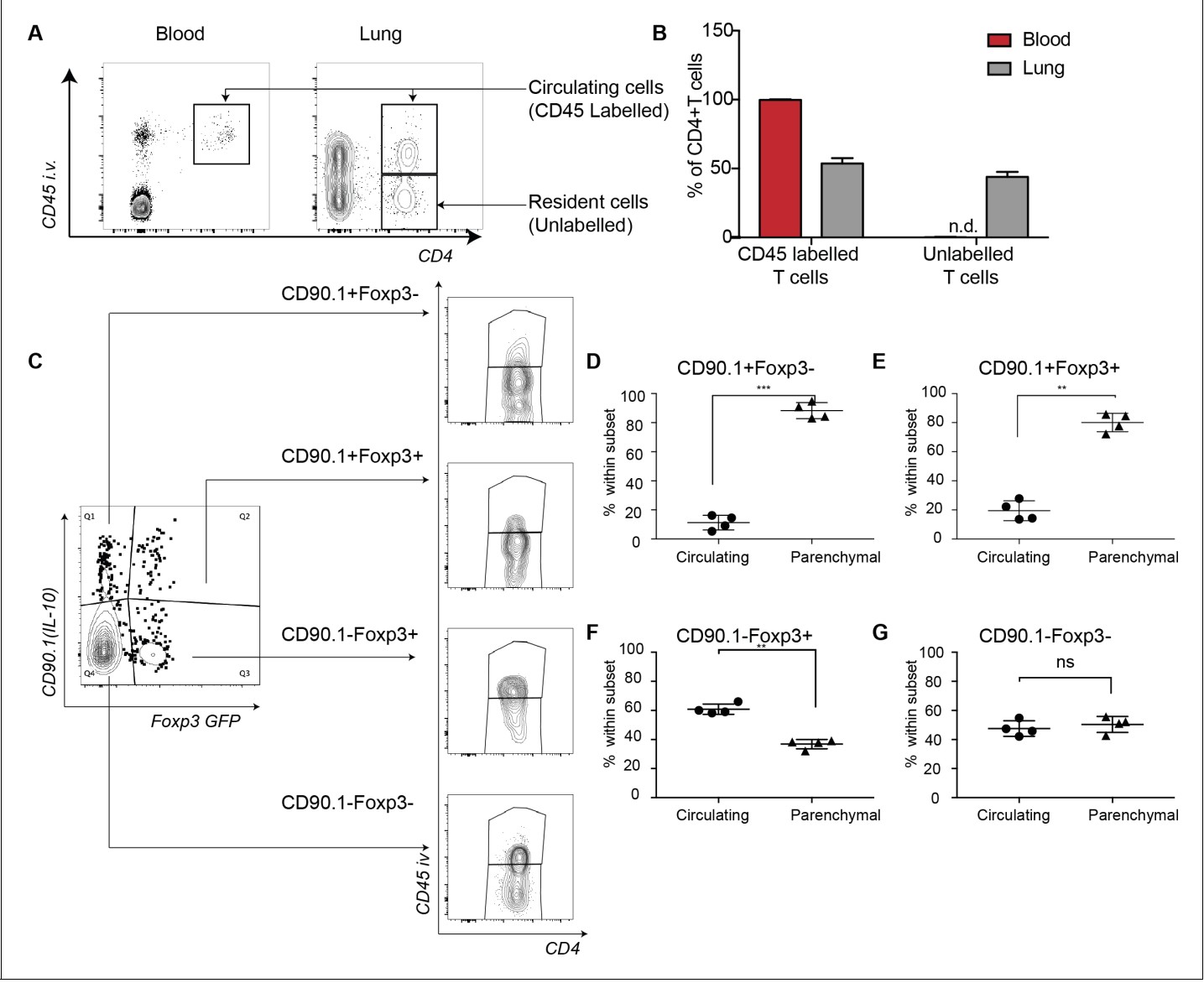

**Figure 2.** IL-10-producing cells are located in the lung parenchyma at peak of inflammation. Intravascular (IV) labeling of cells was performed by retro orbital injection of CD45 antibody and animals were euthanized 2 minutes after injection. Analysis was carried at peak of inflammation day 16-post first house dust mite challenge. (**A**) Efficacy of labeling of cells was assessed in blood and in the lungs. (**B**) The proportion of cells in blood and lung, which were labeled with antibody IV, was quantified. Error bars are standard deviation. Statistical significance was determined using a paired two tailed students t-test. n.d = not detected (**C**) Gating scheme showing the different subsets: CD90.1+ (IL-10+) Foxp3-, CD90.1+ (IL-10+) Foxp3+, CD90.1- (IL-10-) Foxp3+ and CD90.1- (IL-10-) Foxp3- within the CD4 T cells in the lungs. The proportion of circulating CD45+IV labeled cells or tissue-resident (CD45-) within the CD90.1+ (IL-10+) Foxp3-, CD90.1- (IL-10-) Foxp3+ and CD90.1- (IL-10-) Foxp3- is also shown. (**D**) Proportion of cells, which are circulating or resident within the CD90.1+ (IL-10+) Foxp3-, (**E**) within CD90.1- (IL-10-) Foxp3+ and (**F**) within CD90.1- (IL-10-) Foxp3- was quantified. Data are representative of 2 independent experiments**P≤0.01.

DOI: https://doi.org/10.7554/eLife.44821.004

The following source data is available for figure 2:

**Source data 1.** IL-10-producing cells are located in the lung parenchyma at peak of inflammation.

DOI: https://doi.org/10.7554/eLife.44821.005

CD90.1-Foxp3- CD4+ T cells were in the lung vasculature (*Figure 2F–G*). These data indicated that both CD90.1+Foxp3- and CD90.1+Foxp3+ T cells are primarily within the lung parenchyma.

## CD4+Foxp3- T cells are the primary source of IL-10 at the peak of the inflammatory response

Since different cell types can produce IL-10, we further distinguished T cell-derived and non-T cell-derived sources of IL-10 using CD90.1 expression in control and HDM-sensitized animals (*Figure 3A*). We found marked differences in the composition of CD90.1+ cells between control and HDM-sensitized animals. In control animals, non-T cells (CD3-) comprised the majority of CD90.1+ whereas in HDM-sensitized animals, the majority of CD90.1+ were Foxp3-CD4+ T cells (*Figure 3B*). The total number of CD90.1+ Foxp3- cells was predominantly increased in the lungs in comparison to other CD90.1 + subsets (*Figure 3C*). The MFI of CD90.1 as a correlate of IL-10 production was also increased in the CD90.1+Foxp3- subsets as compared to CD3+CD4- and CD3-CD4- subsets (*Figure 3D*). Thus, the increase in the frequency of Tr1-like cells at the site of allergen sensitization could be attributed to the accumulation of CD90.1+ Foxp3- CD4+ T cells in the lungs. These changes were specific to the site of allergen challenge as neither the frequency of total CD90.1+

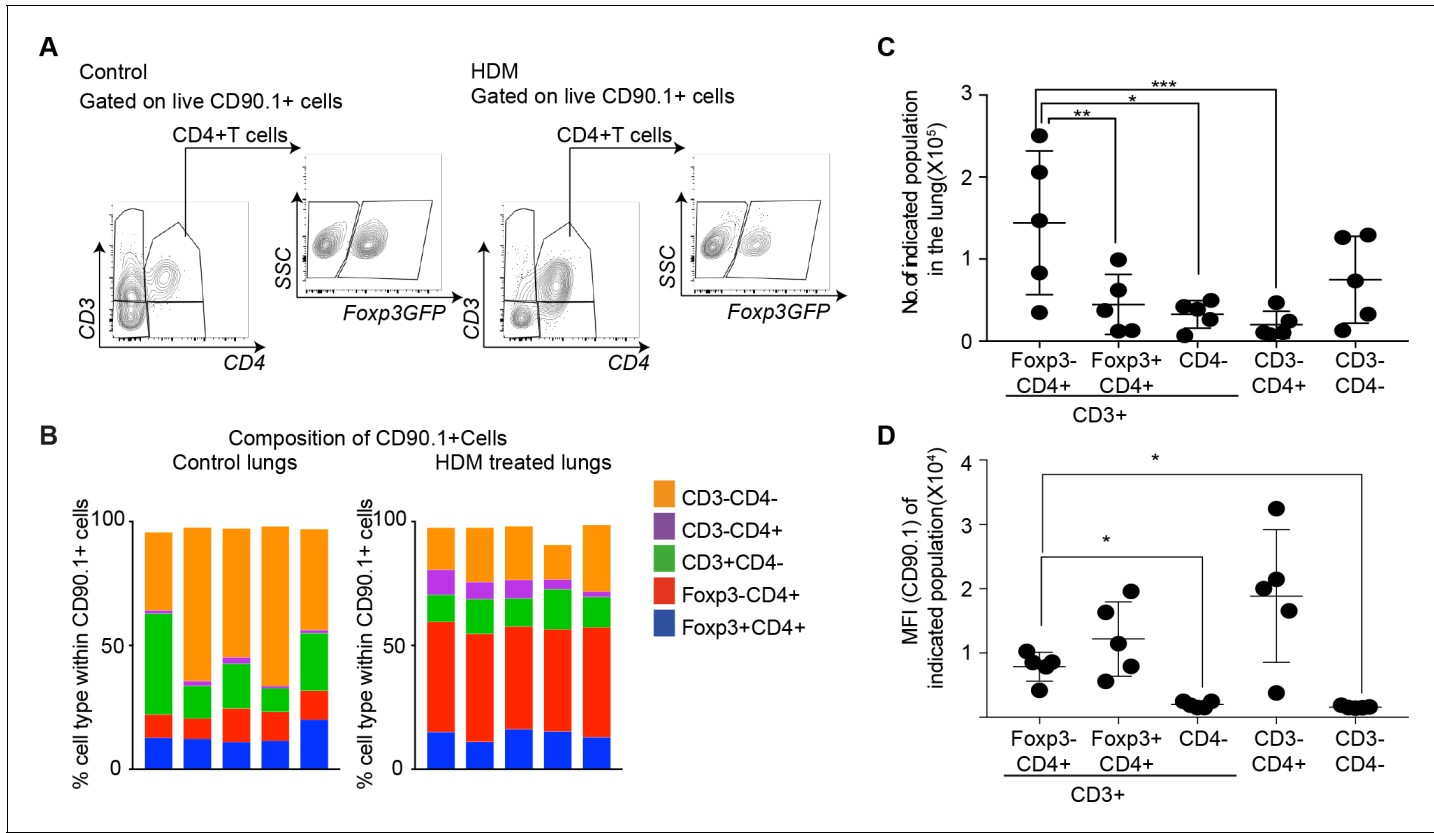

**Figure 3.** CD4+Foxp3- T cells are the prominent IL-10 competent cells at the peak of inflammatory response. (**A**) Gating scheme to identify the composition of IL-10-producing cells in the lungs of control or HDM- treated mice. (**B**) Composition of IL-10-producing cells is shown by plotting the frequency of indicated cell types within all IL-10-producing (CD90.1+) cells in control or HDM-treated lungs. Each column is one animal. n=5 for control group, n=5 for HDM group. (**C**) Total number of indicated subsets in the lungs of HDM-treated animals at peak of inflammation. (**D**) Median fluorescence intensity of CD90.1 in indicated subsets. Statistical significance was determined using one-way ANOVA (post hoc test: Tukey). Data representative of three independent experiments. *P≤0.05, **P≤0.01, ***P≤0.001.
DOI: https://doi.org/10.7554/eLife.44821.006

The following source data and figure supplements are available for figure 3:

**Source data 1.** CD4+Foxp3- T cells are the prominent IL-10 competent cells at the peak of inflammatory response.
DOI: https://doi.org/10.7554/eLife.44821.009

**Figure supplement 1.** Frequency and composition of splenic IL-10-producing cells is comparable in control and HDM-treated mice.
DOI: https://doi.org/10.7554/eLife.44821.007

**Figure supplement 1—source data 1.** Frequency and composition of splenic IL-10-producing cells is comparable in control and HDM-treated mice.
DOI: https://doi.org/10.7554/eLife.44821.008

cells nor their composition was changed in the spleens of sensitized animals (*Figure 3—figure supplement 1A–C*).

## Previously described Tr1 markers do not distinguish IL-10-producing Foxp3- T cells from IL-10-producing Foxp3+ Treg in this model

We next investigated whether CD90.1+ Foxp3- cells in the lungs of HDM-sensitized animals expressed phenotypic markers previously associated with Tr1 cells in comparison to the other CD4+ T cells subsets (*Gagliani et al., 2013*; *Roncarolo et al., 2006*; *Yao et al., 2015*).

We measured the expression of CD25, the alpha subunit of the IL-2 receptor, which is highly expressed on Foxp3+Tregs and correlates with their suppressive function (*Sakaguchi et al., 2009*). CD25 was explicitly increased on Foxp3+ cells irrespective of IL-10 production, though CD90.1 +Foxp3+ exhibited a lower level of expression in comparison to CD90.1-Foxp3+ cells (*Figure 4A*).

We also quantified the expression of KLRG1 and PD1, markers of senescence or exhaustion respectively that correlate with the suppressive function of regulatory T cells (*Burton et al., 2014*; *Shevach, 2006*; *Stephens et al., 2007*). Tr1 cells also express PD-1 during immunotherapy (*Burton et al., 2014*). We found that both KLRG1 and PD-1 are highly expressed on CD90.1+ cells irrespective of Foxp3 expression (*Figure 4B,C*). These differences were reflected in the frequency of cells expressing these markers (*Figure 4D–F*)

All IL-10-producing CD4+ T cells irrespective of Foxp3 expression express high levels of CD44 (*Figure 4G*). CD44 is a receptor for hyaluronan and studies from our group and others have shown that it can potentiate IL-10 responses in CD4+ T cells (*Bollyky et al., 2011*; *Yao et al., 2015*). Despite this association, CD44- /- mice did not show a decrease in the frequency of CD90.1+Foxp3- or CD90.1+Foxp3+ cells in the HDM model, suggesting that CD44 is dispensable for their induction (*Figure 4—figure supplement 1*).

LAG3, an inhibitory immune receptor, and CD49b, an integrin alpha subunit, have been proposed to define Tr1 cells (*Gagliani et al., 2013*). We found that while LAG3 expression distinguished CD90.1+Foxp3- from CD90.1-Foxp3- T cells, it did not distinguish between CD90.1+Foxp3- and either CD90.1+Foxp3+ cells or CD90.1-Foxp3+ cells (*Figure 4H*). CD49b expression was similar between all the different subsets (*Figure 4I*). Similar patterns were also observed for the frequency of cells expressing these markers (*Figure 4J–L*).

Together these data indicate that CD90.1+Foxp3- cells in this model are typically CD44$^{hi}$, CD25$^{lo}$, PD-1$^{hi}$, LAG3$^{hi}$, and KLRG$^{int}$. Thus, these cells share most markers previously associated with Tr1 cells (i.e. they are 'Tr1-like'). However, these markers do not distinguish between Tr1-like cells and other IL-10-producing cells in this model.

## IL-10 production by Tr1-like cells is transient and wanes after the peak of inflammation

We next sought to determine the kinetics of IL-10 production and airway inflammation in this model. To this end, we analyzed responses at day 2, day 6, day 16 and day 30 after allergic sensitization. For these experiments, the same protocol was used as in *Figure 1* but with additional analysis timepoints; a schematic of these is shown in *Figure 5A*.

The inflammatory response peaked on day 16 as measured by total cellular (*Figure 5B*) and eosinophilic infiltration (*Figure 5C*) in the BAL. By day 30, inflammation was largely resolved. The frequency of CD90.1+Foxp3- (Tr1-like cells) within the CD4+ T cell population was likewise maximal at day 16 and waned by day 30 (*Figure 5D,E*). The frequency of CD90.1+Foxp3+ Treg was also maximal at day 16 (*Figure 5F*), while the frequency of CD90.1- Foxp3+ cells was unchanged (data not shown). Similarly, the frequency of CD90.1+ Foxp3- T cells also diminished in the BAL from Day 16 to day 30, whereas the CD90.1+Foxp3+ and CD90.1-Foxp3+ cells were unchanged (*Figure 5—figure supplement 1*)

To directly measure IL-10 production independently of CD90.1, we performed intracellular cytokine staining on these cells' ex vivo. At the peak of inflammation on day 16 the majority of CD90.1+ Foxp3- cells produced IL-10 and insignificant amounts of IFNg, IL-13, or IL-17 in the lungs (*Figure 5G*) and in the BAL (*Figure 5—figure supplement 1*). However, by day 30 IL-10 production by CD90.1 cells had waned (*Figure 5G*).

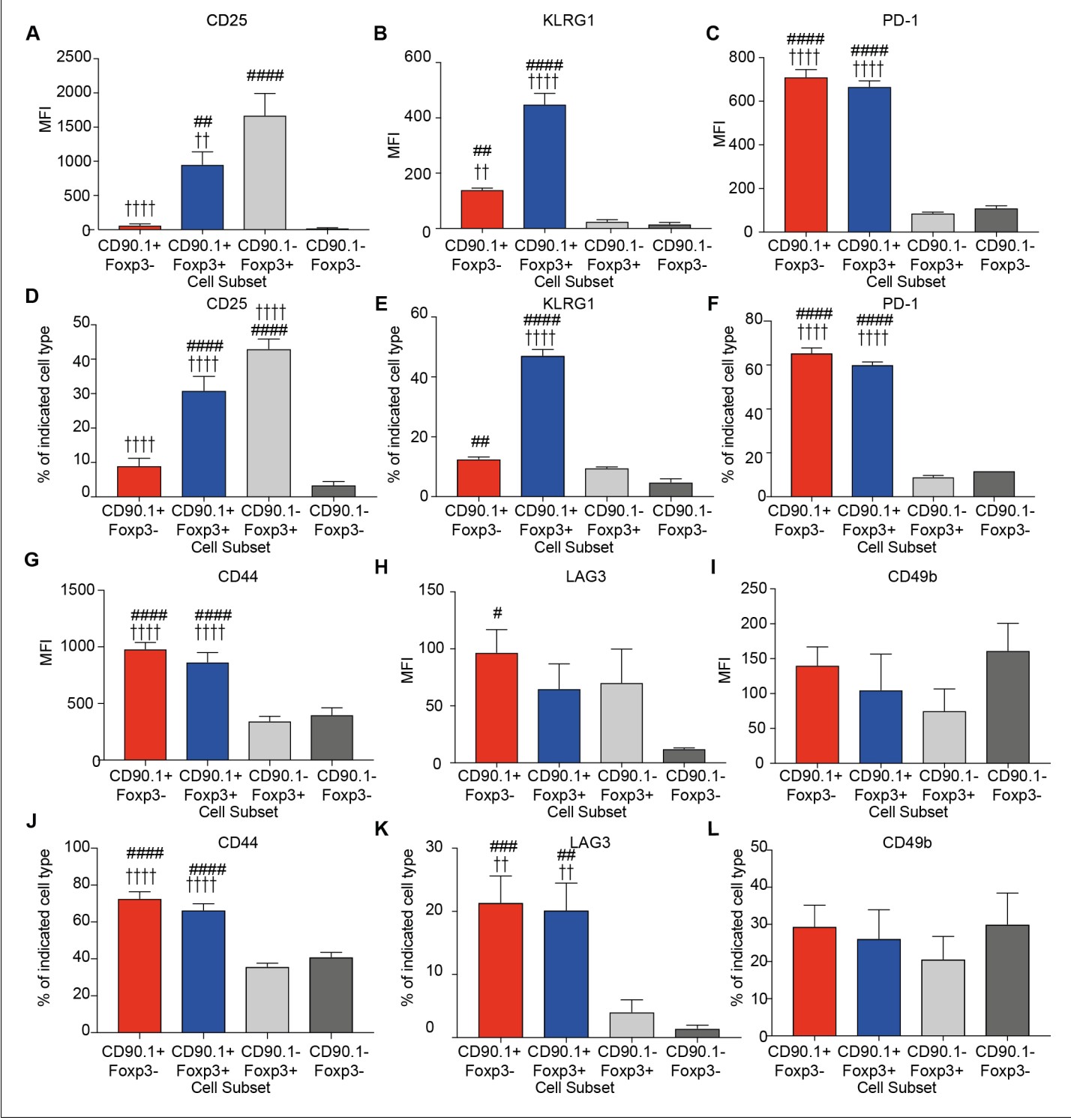

**Figure 4.** Phenotype of lung CD4 T cell subsets at the peak of inflammation. The median fluorescence intensity of (A) CD25, (B) KLRG1, (C) PD-1, and percentage of (D) CD25, (E) KLRG1, (F) PD-1 and MFI of (G) CD44, (H) LAG3, (I) CD49b and percentage of (J) CD44, (K) LAG3 and (L) CD49b in indicated CD4+ T cell subsets in the lungs of HDM-treated mice 4 days after last challenge. Data are pooled from 2 experiments. Error bars represent standard error of mean. n=9-10 for control group, n=9-10 for HDM group for CD44, LAG3, CD49b and CD25. n=5 for control group, n=5 for HDM group for KLRG-1 and PD-1. Statistical significance was determined using 2-way ANOVA (post hoc test: Bonferroni). Data are representative of three independent experiments. # represents significant difference from CD90.1-Foxp3- cells and represents significant difference from CD90.1-Foxp3+ cells. #P≤0.05, ## P≤0.01, P≤0.01, ####P≤0.001 P≤0.001.

*Figure 4 continued on next page*

*Figure 4 continued*

DOI: https://doi.org/10.7554/eLife.44821.010

The following source data and figure supplements are available for figure 4:

**Source data 1.** Phenotype of lung CD4 T cell subsets at the peak of inflammation.

DOI: https://doi.org/10.7554/eLife.44821.013

**Figure supplement 1.** IL-10-producing CD4+ T cells do not require CD44.

DOI: https://doi.org/10.7554/eLife.44821.011

**Figure supplement 1—source data 1.** IL-10-producing CD4+ T cells do not require CD44.

DOI: https://doi.org/10.7554/eLife.44821.012

These CD90.1+IL-10- 'former Tr1' remained within the lung parenchyma as assessed by intravenous labeling (*Figure 5H*). Moreover, majority of these cells were CD62L negative and CD44 high hence exhibiting an effector memory phenotype (*Figure 5I*).

Together, these data indicate that Tr1-like cells and not Foxp3+ Treg comprise the majority of IL-10+ T cells in this model. Further, these data suggest that some CD90.1+cells may contribute to the allergen specific memory T-cell pool.

## Active IL-10 production in Tr1 like cells is associated with *Irf1* and *Batf* expression

Given the low levels of IL-10 production in CD90.1+ cells 30 days after antigenic challenge (*Figure 5G*), we questioned whether CD90.1+ cells require persistent antigenic signals for active IL-10 production. To address this, we isolated CD90.1+ cells from spleens of 10BiT mice and cultured them with or without anti-CD3 and anti-CD28 as described previously (*Chihara et al., 2016*) Only cells which were activated continued to produce IL-10 after 5 days in cell culture (*Figure 6A,B*). Moreover, the viability of cultured cells was severely affected in the absence of TCR stimulation over time (*Figure 6—figure supplement 1*).

We then asked whether expression of *Irf1* and *Batf*; transcription factors associated with Tr1 cells and IL-10 production (*Karwacz et al., 2017*), distinguished CD90.1+IL-10+ Tr1-like cells from CD90.1+IL-10- 'former' Tr1-like cells in an activation dependent manner. Using antibodies against CD90.1 and IL-10 cytokine capture assays we sorted CD90.1+IL-10+, CD90.1+IL-10-, and CD90.1-IL-10- cell populations, using the latter for normalization.

We found that *Irf1* and *Batf* are upregulated only in cells actively producing IL-10 (*Figure 6C,D*). These transcription factors therefore characterize active IL-10 production by Tr1-like cells in this model.

## Tr1-like cells are a part of allergen-specific memory in the lungs of previously sensitized animals

We next investigated the role of previous allergen exposure in IL-10 production. For this we established a long-term model of HDM challenge (*Figure 7A*). In brief, we sensitized animals to HDM over 2 weeks as before. Following this, the animals were rested for close to 2 months. Then, on day 67 after the first challenge, both PBS and HDM-sensitized animals were challenged with one dose of HDM, and the airway inflammatory response was analyzed 24 hr later. Looking this early after allergen challenge allowed us to better assess the rapid CD4 T cell allergen specific memory responses.

Using this protocol, we found that animals that received primary sensitization with HDM exhibited heightened airway inflammation (*Figure 7B*), increased eosinophils (*Figure 7C*), and heightened IgE upon rechallenge (*Figure 7D*). CD90.1+Foxp3- cells were increased in HDM-sensitized animals (*Figure 7E*) while the frequency of CD90.1+Foxp3+ as well as CD90.1-Foxp3+ cells were unchanged. Consequently, the frequency of Foxp3-CD4+ T cells within all IL-10-producing cells was also elevated in HDM-sensitized animals in comparison to control, although non-T cells remained the main producers of IL-10 in both groups (*Figure 7F*). These changes were specific to the lung as we did not observe any changes in these subsets in the draining lymph node nor spleens (not shown).

To interrogate the antigen specificity of these cells, we used an I-A[b] tetramer containing epitope 117–124 of the Derp1 protein from HDM (*Hondowicz et al., 2016*). Most tetramer + cells were either CD90.1+Foxp3- or CD90.1-Foxp3- while all Foxp3+ cells irrespective of CD90.1 expression

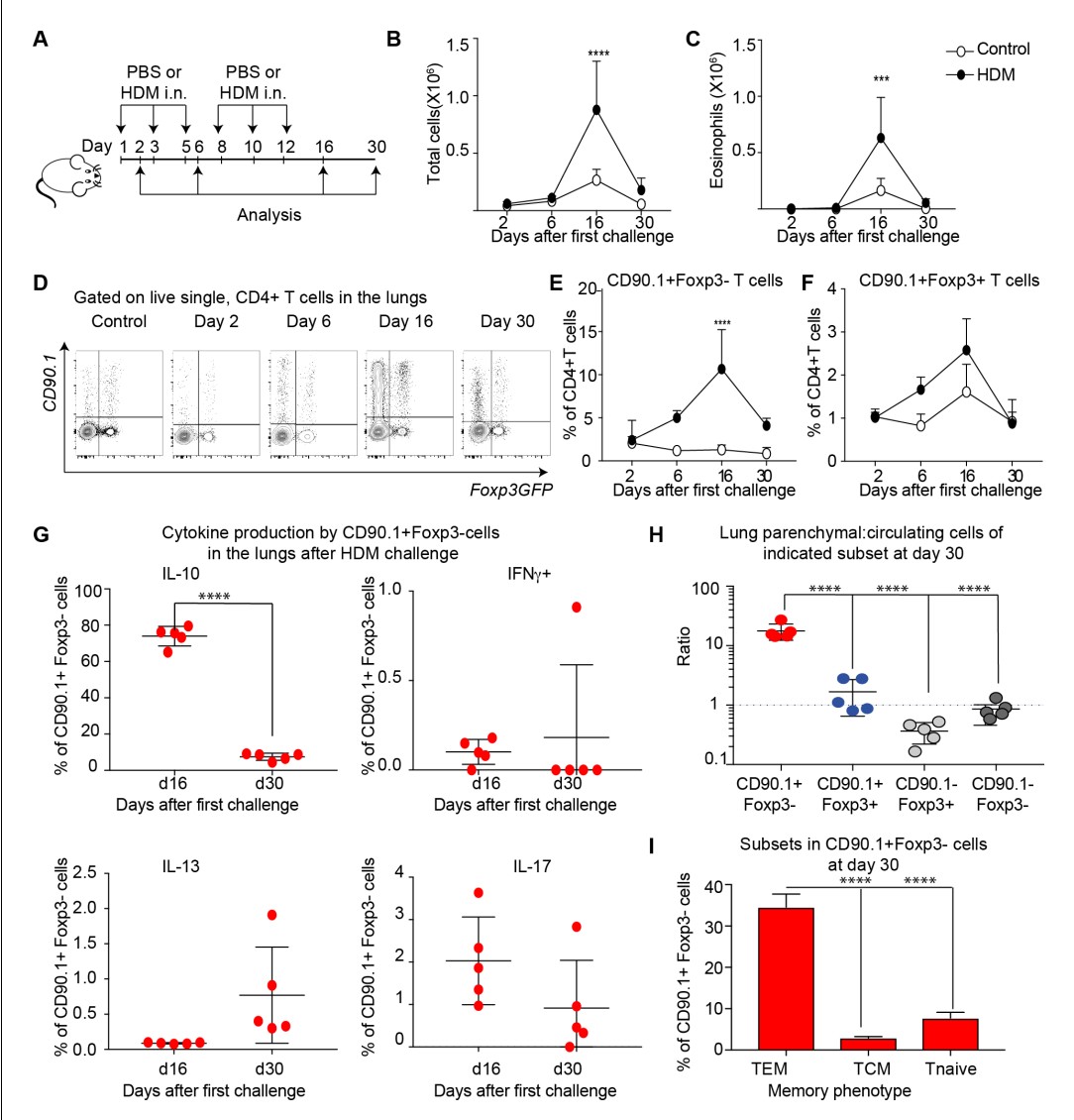

**Figure 5.** IL-10 production by Tr1-like cells is transient and wanes after peak of inflammation. (A) 10BiT reporter mice on a C57Bl/6 background were administered PBS as control or HDM and analysis was performed on groups of mice at the indicated times shown. (B) The total number of cells and (C) total number of eosinophils in the BAL were determined. n=3-5 animals per group per time point. Error bars represent standard deviation of the mean. (D) Representative flow cytometry plots showing the kinetics of IL-10-producing T cell subsets as identified by surface expression of CD90.1 and Foxp3+ regulatory T cell subsets as identified by GFP expression. The frequency of (E) IL-10-producing Foxp3- (CD90.1+Foxp3-), (F) IL-10-producing Foxp3+ (CD90.1+Foxp3+) within all CD4+ T cells is plotted at indicated analysis days. n=3-5 animals per group per time point. Error bars represent standard deviation of the mean. Statistical significance was determined using 2-way ANOVA (post hoc test: Sidaks). (G) Cells from lungs of HDM- treated mice obtained after Day 26 and day 30 were stimulated with PMA, Ionomycin and Brefeldin and surface stained, fixed and permeabilized for detection of cytokines. The frequency of CD90.1+Foxp3- T cells producing IL-10, IFNγ, IL-13 and IL-17 is shown for indicated time points. (H) To identify lung resident T cells, intravascular (IV) labeling of cells was performed by retroorbital injection of CD45 antibody and mice were euthanized 2 minutes after injection. The ratio of resident to circulating cells within indicated CD4+ T cells from lungs of HDM-treated mice obtained at day 30 post first challenge is plotted. (I) The frequency of effector (TEM) (CD62l-CD44+), central (TCM) (CD62l+CD44+), and naïve subsets(CD62L+CD44-) within CD90.1+Foxp3- cells in lungs at day 30 post first allergen challenge is plotted. Data representative of three independent experiments ***P≤0.001, ****P≤0.0001 GFP= green fluorescent protein. PMA= phorbol myristate acetate, IFNγ= Interferon gamma.

DOI: https://doi.org/10.7554/eLife.44821.014

The following source data and figure supplements are available for figure 5:

**Source data 1.** IL-10 production by Tr1-like cells is transient and wanes after peak of inflammation.
DOI: https://doi.org/10.7554/eLife.44821.017

**Figure supplement 1.** CD90.1+Foxp3-, CD90.1+Foxp3+ and CD90.1-Foxp3- cells frequency and cytokine production in the BAL.

*Figure 5 continued on next page*

*Figure 5 continued*

DOI: https://doi.org/10.7554/eLife.44821.015

**Figure supplement 1—source data 1.** CD90.1+Foxp3-, CD90.1+Foxp3+ and CD90.1-Foxp3- cells frequency and cytokine production in the BAL.
DOI: https://doi.org/10.7554/eLife.44821.016

were negative for tetramer staining (*Figure 7G*, *Figure 7—figure supplement 1*). The population of tetramer-positive Tr1-like cells increased upon antigenic challenge while tetramer-staining Foxp3+ Treg populations did not. Within CD90.1+Foxp3-CD4+ T cells, a greater frequency of cells were tetramer+ (*Figure 7H*).

To further define the functional phenotype of these CD90.1+ Foxp3- T cells, we measured cytokine production after ex vivo stimulation. We found that the majority of CD90.1+Foxp3- cells produced IL-10 and fewer cells produced other cytokines such as IL-4, IL-13, IFNg and IL-17 (*Figure 7I*, *Figure 7—figure supplement 2*). Additionally, we found that these Tr1-like cells also expressed low levels of GATA3 and higher levels of Tbet (*Figure 7J*, *Figure 7—figure supplement 2*).

Together, these data indicate that Tr1-like cells arise from the allergen-specific memory response in this model. Further, the tetramer-positive cells that produced IL-10 in this model were Tr1-like cells and not Foxp3+ Treg.

## IL-10-producing T cells in the lung can arise from tissue resident memory cells

We next asked whether the Tr1-like cells found upon memory challenge are resident to the lung or circulating. Using the same long-term model of HDM challenge used in *Figure 6A*, we observed that most CD90.1+ cells exhibited an effector memory phenotype (CD44hi, CD62Llow) as compared to the CD90.1-Foxp3- cells (*Figure 8A*). In addition, using intravascular labeling, we found that CD90.1+Foxp3- cells had a higher ratio of resident to circulating T cells (*Figure 8B*).

To functionally address tissue residency of these cells, we treated mice sensitized to HDM with the S1P receptor agonist FTY720 to block T cell egress from lymph nodes prior to memory rechallenge (*Figure 8C*, *Figure 8—figure supplement 1*). We found that disruption of lymphocyte egress

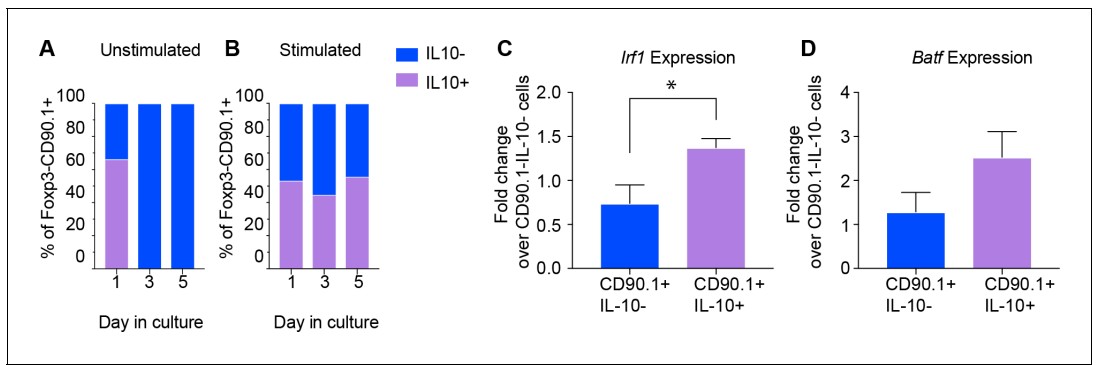

**Figure 6.** Active IL-10 production is associated with *Irf1* and *Batf* expression. CD90.1- and CD90.1+ CD4 T cells were isolated from 10BiT spleens and cultured (**A**) unstimulated in plain media or (**B**) with CD3/CD28 stimulation for 5 days to assess kinetics of Thy1.1 surface expression and intracellular IL-10 cytokine staining. (**C**) *Irf1* expression and (**D**) *Batf* expression in indicated subsets obtained from in vitro differentiated Tr1 cell cultures. Data was normalized to beta actin as reference gene and is expressed as fold change over 90.1-IL-10- cells using delta $C_t$ method. Expression data are pooled from five independent experiments.
DOI: https://doi.org/10.7554/eLife.44821.018

The following source data and figure supplements are available for figure 6:

**Source data 1.** Active IL-10 production is associated with *Irf1* and *Batf* expression.
DOI: https://doi.org/10.7554/eLife.44821.021

**Figure supplement 1.** Viability of ex vivo Tr1 cells with and without TCR stimulation.
DOI: https://doi.org/10.7554/eLife.44821.019

**Figure supplement 1—source data 1.** Viability of ex vivo Tr1 cells with and without TCR stimulation.
DOI: https://doi.org/10.7554/eLife.44821.020

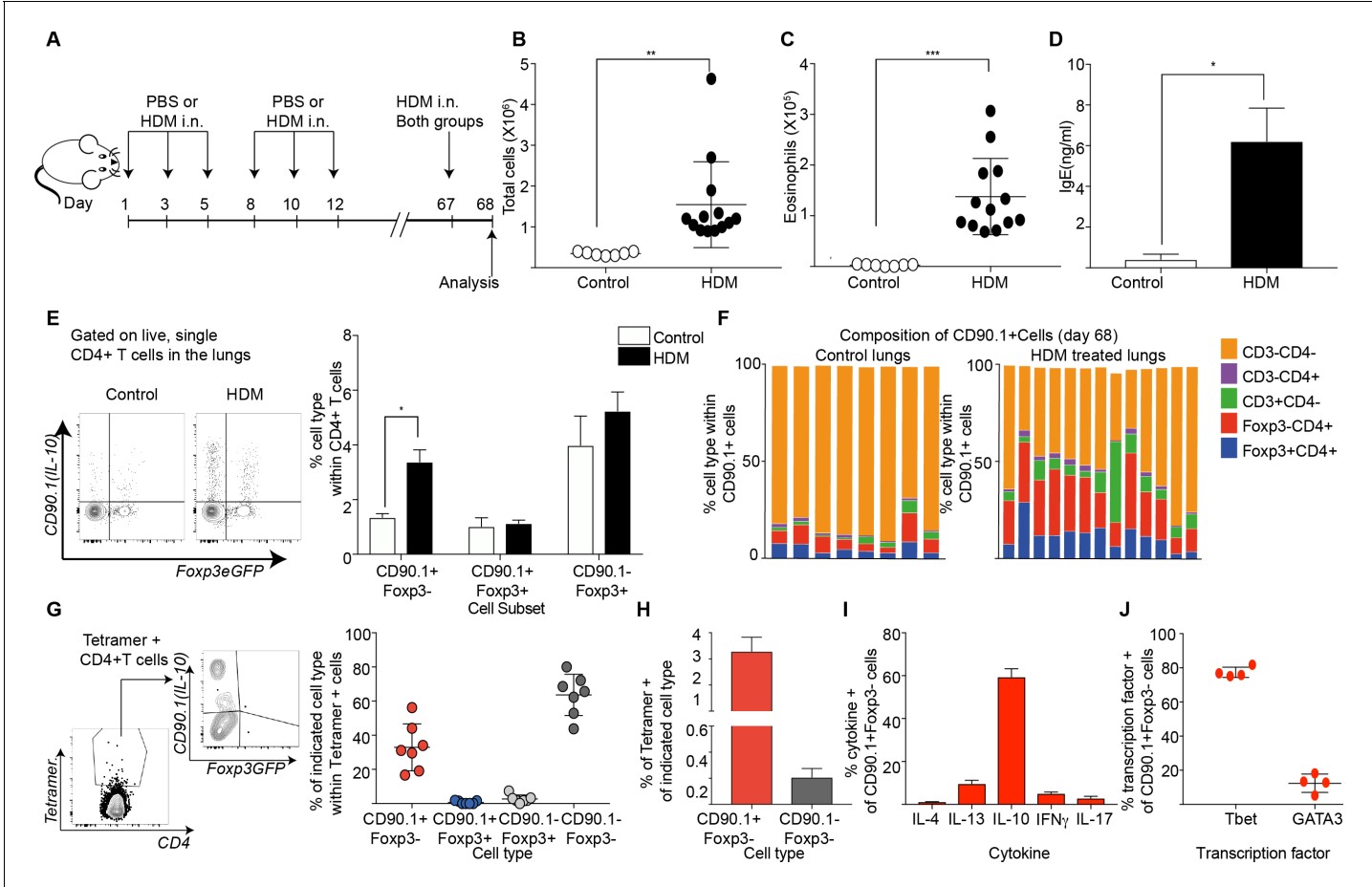

**Figure 7.** Tr1-like cells contribute to allergen-specific memory T-cells in the lung. (**A**) Mice were administered either PBS as control or crude HDM intranasally (i.n.) Six times over 2 weeks as shown. The mice were then left unchallenged until day 67 when both PBS (control) and HDM-sensitized animals were challenged with HDM. Analysis was done one day after the rechallenge at day 68. (**B**) The total number of cells in the BAL was determined. Data are pooled from two experiments. Each symbol represents a single animal. Error bars represent standard error of mean. n=7 for control group, n=13 for HDM group. Statistical significance was determined using an unpaired two tailed students t test. (**C**) The total number of eosinophils in the BAL is also plotted. Data are pooled from two experiments. Each symbol represents a single animal. Error bars represent standard error of mean. n=7 for control group, n=13 for HDM group. Statistical significance was determined using an unpaired two tailed students t test. **P≤0.01, ***P≤0.001. (**D**) The level of IgE in BAL supernatant was determined by ELISA. n=8 for control group n=16 for HDM group. (**E**) Representative flow cytometry plot showing all IL-10-producing cells within CD4+ T cells in lungs of control or HDM-treated animals at day 68. (**F**) Composition of IL-10-producing cells is shown by plotting the frequency of indicated cell types within all IL-10-producing (CD90.1+) cells. Data are pooled from two experiments. Each column is an individual animal. (**G**) Gating scheme showing the HDM specific T cells stained using Derp1 tetramer and the different T cell subsets that constitute the tetramer + CD4+ T cells in the lungs of HDM-treated animals. The frequency of IL-10-producing Foxp3- (CD90.1 +Foxp3-), IL-10-producing Foxp3+ (CD90.1+Foxp3+), Foxp3+ cells which do not produce IL-10 (CD90.1-Foxp3+) and CD90.1-Foxp3- within all CD4+ T cells is plotted. (**H**) The frequency of Tetramer + cells within CD90.1-Foxp3- and CD90.1+Foxp3- cells in the lungs of HDM-treated animals is plotted. Error bars represent standard deviation of the mean. n=7. (**I**) PMA, Ionomycin and Brefeldin stimulated CD4 T cells from lungs of HDM-treated mice were surface stained, fixed and permeabilized for detection of cytokines. The frequency of cells producing IL-4, IL-13, IL-10, IFNγ and IL-17 within CD90.1+ Foxp3- (Tr1-like) CD4+ T cells is shown. Data are pooled from two experiments. N is between 4-13 mice per cytokine (**J**). The frequency of cells expressing transcription factors Tbet and GATA3 the within CD90.1+ Foxp3- (Tr1-like) CD4+ T cells is shown. Error bars represent standard deviation of mean. n=13. Data representative of three independent experiments.

DOI: https://doi.org/10.7554/eLife.44821.022

The following source data and figure supplements are available for figure 7:

**Source data 1.** Tr1-like cells contribute to allergen-specific memory T-cells in the lung.
DOI: https://doi.org/10.7554/eLife.44821.027
**Figure supplement 1.** Gating scheme showing Tetramer positive cells in (**A**) control and (**B**) HDM-treated lungs after memory challenge.
DOI: https://doi.org/10.7554/eLife.44821.023
**Figure supplement 1—source data 1.** Tetramer positive cells in control and HDM-treated lungs after memory challenge.

*Figure 7 continued on next page*

*Figure 7 continued*

DOI: https://doi.org/10.7554/eLife.44821.024

**Figure supplement 2.** Phenotype of CD4 subsets during memory rechallenge, gated on CD90.1 and Foxp3 expression.

DOI: https://doi.org/10.7554/eLife.44821.025

**Figure supplement 2—source data 1.** Phenotype of CD4 subsets during memory rechallenge, gated on CD90.1 and Foxp3 expression.

DOI: https://doi.org/10.7554/eLife.44821.026

did not affect cellular infiltration or eosinophilia in the BAL (*Figure 8D,E*). However, the ratio of CD62L+ to CD62L- CD4+ T cells in the lungs was significantly reduced (*Figure 8F*). This is in line with a previous report that lung resident CD62L negative memory cells are sufficient to induce airway inflammation (*Hondowicz et al., 2016*). Notably, the frequency of Tr1-like cells within the CD4 +T cells in the lungs was unaffected (*Figure 8G*). Unlike CD90.1-Foxp3- T cells, most of the CD90.1 +Foxp3- Tr1-like cells were consistently effector memory (CD44hi CD62L low) both with and without FTY270 administration (*Figure 8H,I*).

To address the memory phenotype of our CD90.1+Foxp3- CD4 T cells in the lung parenchyma, we analyzed lungs from memory mice on day 68 without allergen rechallenge looking for common markers of tissue-residency (*Beura and Masopust, 2014*; *Schenkel and Masopust, 2014*). We find that when compared to unsensitized mice, CD90.1+Foxp3- cells in HDM-sensitized mice show higher frequencies of CD69 expression and lower frequencies of Ly6C expression. There was no difference in CCR7, CD103, IL7Ra or KI67 expression in CD90.1+Foxp3- cells between sensitization conditions. CD90.1-Foxp3- cells showed no difference in expression of any of the markers analyzed between sensitization conditions. (*Figure 8—figure supplement 2*).

Together, these data indicate that the majority of CD90.1+Foxp3- CD4+ Tr1-like cells following allergen challenge have a memory phenotype and can arise from lung-resident memory cells.

## Neither depletion nor adoptive transfer of Tr1-like cells influences tolerogenic responses to allergen re-challenge

To elucidate the functional contribution of Tr1-like cells to allergic inflammation, we next examined their function using depletion experiments. To this end we administered an antibody directed against CD90.1 to deplete IL-10 producing cells, as has been done previously to deplete CD90.1+ cells in this model (*Maynard et al., 2007*). The strategy was likewise successful here in depleting all CD90.1+ cells including CD90.1+Foxp3- and CD90.1+Foxp3+ T cells from the lungs (*Figure 9A,B*). Importantly the antibody clone used for depleting these cells did not mask CD90.1 staining with other clones (*Figure 9—figure supplement 1*). In addition, it depleted CD90.1+CD3- cells however, IL-10 production by them was unaffected (*Figure 9—figure supplement 2*).

We first depleted the CD90.1+ cell population during the sensitization phase (*Figure 9A,B*). This exacerbated airway inflammation as measured by total cell and eosinophil infiltration in the BAL (*Figure 9C,D*) as well as IL-13 production by CD90.1-Foxp3- CD4+T cells (*Figure 9E*).

We next depleted the CD90.1+ cell population during the memory phase (*Figure 9F,G*). This did not impact the inflammatory response (*Figure 9H,I*), cytokine production by CD90.1-Foxp3- cells (*Figure 9J*), nor the frequency of tetramer + cells within CD4+T cells in the lungs (*Figure 9K*).

We also performed intra-tracheal adoptive transfer of memory CD90.1+ Tr1-like cells or CD90.1- T effector cells from lungs of memory rechallenged mice sensitized to HDM or an unrelated allergen (ovalbumin) into naïve recipients who were subsequently challenged with HDM. Consistent with our observations in the depletion studies, we did not see any beneficial impact of the transfer of memory Tr1-like cells on cellular infiltration or eosinophilia in the BAL (*Figure 10A–C*). Moreover, the transferred CD90.1+ T cells were not more likely than transferred CD90.1- T cells to produce IL-10 upon rechallenge with allergen (*Figure 10D*) and did not alter the levels of IL-10 in the lung (*Figure 10E*). In contrast, CD90.1+Foxp3- cells could suppress cell infiltration in the BAL when transferred in an acute model and were also suppressive in vitro (*Figure 10—figure supplement 1*). This suggests that CD90.1+Foxp3- cells are unable to suppress the allergen specific inflammatory memory responses. An inherent problem of this model is that neither CD90.1+ nor CD90.1 negative cells engraft efficiently in the lungs (*Figure 10—figure supplement 2*).

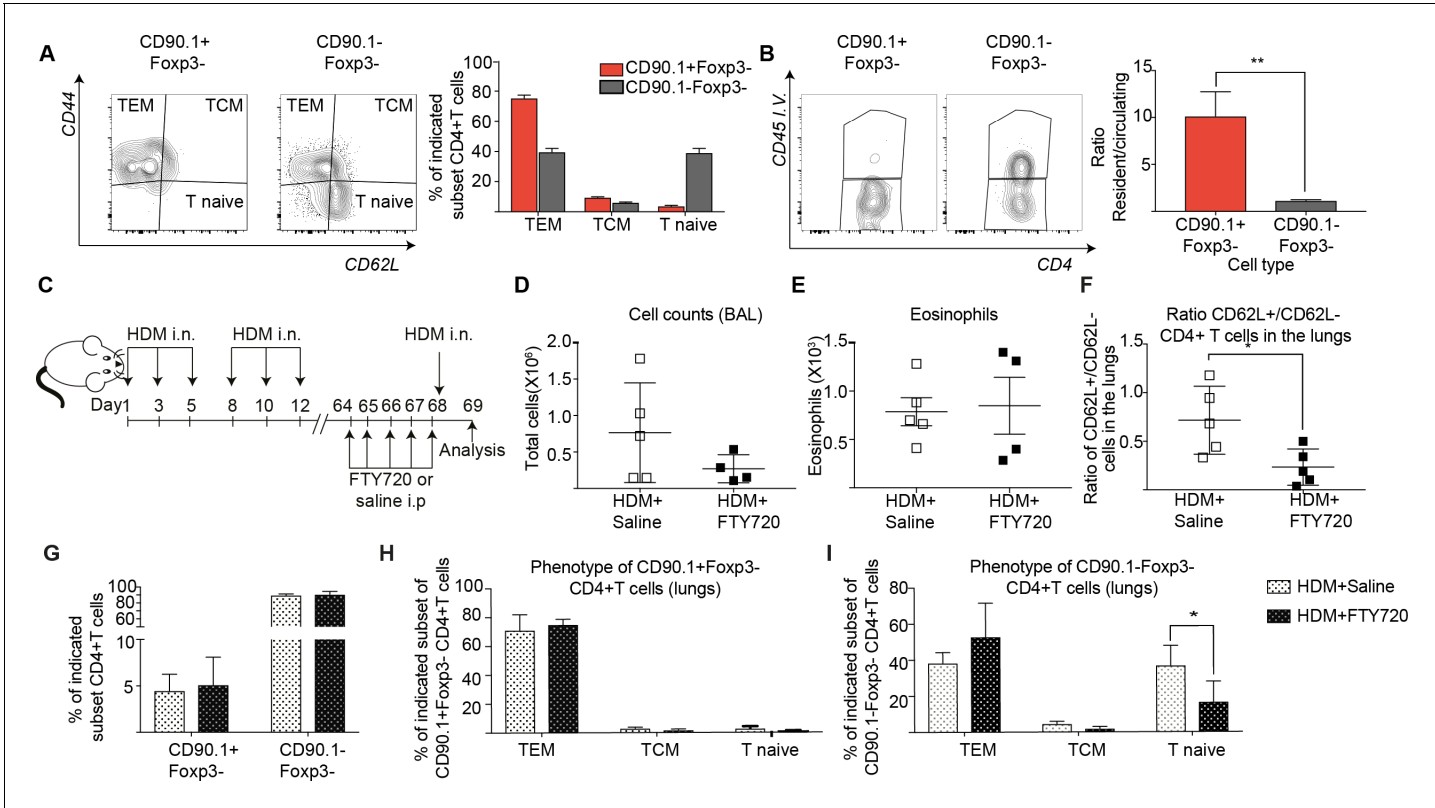

**Figure 8.** IL-10-producing T cells in the lung can originate from tissue resident memory cells. (**A**) Gating scheme showing different memory cell subsets of Tr1-like (CD90.1+Foxp3-) or (CD90.1-Foxp3-) CD4+ T cells in the lungs of mice treated with HDM using the same protocol outlined in Figure 6A. The frequency of each memory subset within CD90.1+Foxp3- CD4+ T cells or CD90.1-Foxp3- T cells in the lungs of HDM-treated mice is plotted on the left. Data are pooled from two experiments. Error bars represent standard error of mean. n=10 for control group, n=10 for HDM group. To identify lung resident T cells, intravascular (IV) labeling of cells was performed by retroorbital injection of CD45 antibody and mice were euthanized 2 minutes after injection. (**B**) Representative flow cytometry plot showing resident versus circulating T cells within CD90.1+Foxp3- and CD90.1-Foxp3- CD4+ T cells from lungs of HDM-treated mice is shown. The ratio of resident to circulating cells within CD90.1+Foxp3- and CD90.1-Foxp3- CD4+ T cells from lungs of HDM-treated mice is plotted. Data are pooled from two experiments. Error bars represent standard error of mean. n=9 for control group, n=9 for HDM group. Statistical significance was determined using an unpaired two tailed students t test. Data representative of three independent experiments (**C**) To block lymphocyte egress from lymph nodes, HDM-sensitized animals were treated with FTY720. (**D**) The total number of cells in the BAL was determined. Each symbol represents a single animal (**E**) The total number of eosinophils in the BAL is also plotted. Each symbol represents a single animal. (**F**) The ratio of CD62L+/CD62L- CD4+ T cells in the lungs is plotted. (**G**) The frequency of CD90.1+Foxp3- CD4+ (Tr1-like T cells) and CD90.1-Foxp3- CD4+ (conventional T cells) in the lungs of control or FTY720-treated mice is shown. (**H**) Frequency of different memory subsets within CD90.1+Foxp3- (Tr1-like cells) and (**I**) CD90.1-Foxp3- (T conventional) cells in the lungs of control or FTY720-treated animals is plotted. n=5 for each group. **P≤0.01 TEM=Effector memory TCM =T central memory.

DOI: https://doi.org/10.7554/eLife.44821.028

The following source data and figure supplements are available for figure 8:

**Source data 1.** IL-10-producing T cells in the lung can originate from tissue resident memory cells.
DOI: https://doi.org/10.7554/eLife.44821.033
**Figure supplement 1.** Efficiency of FTY270 treatment.
DOI: https://doi.org/10.7554/eLife.44821.029
**Figure supplement 1—source data 1.** Efficiency of FTY270 treatment.
DOI: https://doi.org/10.7554/eLife.44821.030
**Figure supplement 2.** long-term persistence of CD90.1+ cells in allergen sensitized lungs.
DOI: https://doi.org/10.7554/eLife.44821.031
**Figure supplement 2—source data 1.** long-term persistence of CD90.1+ cells in allergen sensitized lungs.
DOI: https://doi.org/10.7554/eLife.44821.032

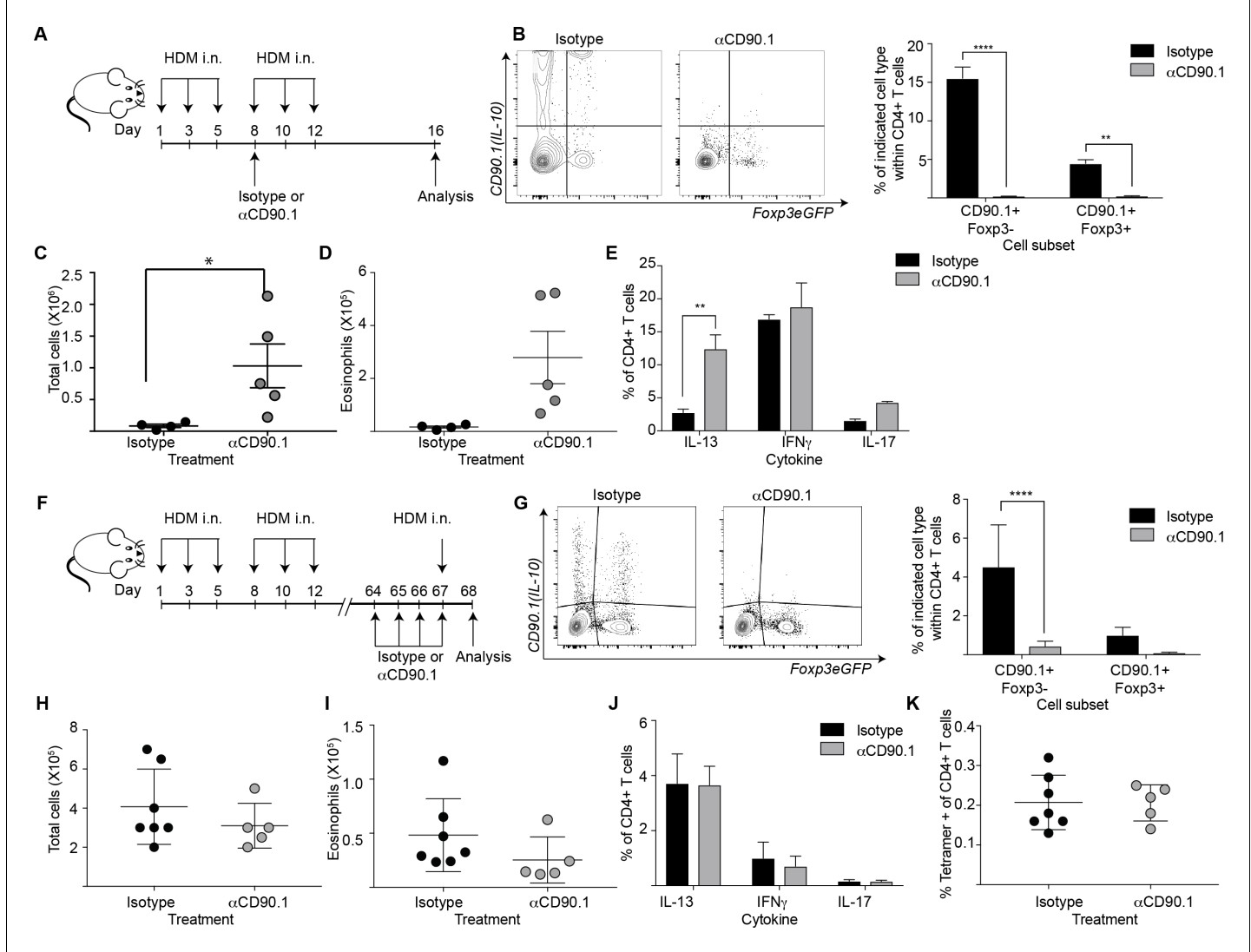

**Figure 9.** Depletion of CD90.1+Foxp3- IL-10 competent Tr1 cells does not influence long-term tolerance to airway allergens. (**A**) Schematic illustrating the protocol used in experiments A-E. CD90.1+ cells were depleted in the sensitization phase using an anti CD90.1 antibody as depicted. Control animals were given matched isotype. (**B**) Representative flow cytometry plots showing the efficiency of depletion of CD90.1+ CD4 T cells in the lungs. Frequency of CD90.1+Foxp3- and CD90.1+Foxp3+ cells within CD4+ T cells in lungs of isotype or anti CD90.1 treated mice are plotted. Error bars represent standard deviation of the mean. n = 4 for isotype group, n = 5 for anti-CD90.1 group. Statistical significance was determined using 2-way ANOVA (post hoc test: Sidaks). (**C**) The total number of cells and (**D**) total number of eosinophils in the BAL was determined. Each symbol represents a single animal. Error bars represent standard deviation of mean. (**E**) Lung CD4 cells from isotype or anti-CD90.1 treated mice were restimulated with PMA, Ionomycin and Brefeldin A, surface stained, fixed and permeabilized for detection of intracellular cytokines. The frequency of cells producing IL-13, IL-17, and IFNγ within Foxp3- CD90.1- CD4+ T cells are shown. Error bars represent standard deviation of mean. n = 4 for isotype group, n = 5 for anti CD90.1 group. (**F**) Schematic illustrating the protocol used in experiments F-K. CD90.1+ cells were depleted using an anti CD90.1 antibody as depicted. Control animals were given matched isotype. (**G**) Representative flow cytometry plots showing the efficiency of depletion of CD90.1+ CD4 T cells in the lungs. Frequency of CD90.1+Foxp3- and CD90.1+Foxp3+ cells within CD4+ T cells in lungs of isotype or anti CD90.1 treated mice are plotted. Error bars represent standard deviation of the mean. n = 7 for isotype group, n = 5 for anti CD90.1 group. Statistical significance was determined using 2-way ANOVA (post hoc test: Sidaks). (**H**) The total number of cells and (**I**) total number of eosinophils in the BAL was determined. Each symbol represents a single animal. Error bars represent standard deviation of mean. (**J**) Lung CD4 cells from isotype or anti-CD90.1 treated mice were restimulated with PMA, Ionomycin and Brefeldin A, surface stained, fixed and permeabilized for detection of cytokines. The frequency of cells producing IL-13, IL-17, and IFNγ within Foxp3- CD90.1- CD4+ T cells are shown. Error bars represent standard deviation of mean. n = 7 for isotype group, n = 5 for anti CD90.1 group. (**K**) The frequency of Tetramer + cells within CD90.1-Foxp3- and CD90.1+Foxp3- cells in the lungs of HDM-treated animals is plotted. Error bars represent standard deviation of the mean. n = 7 for isotype group, n = 5 for anti CD90.1 group. Data representative of at least two independent experiments.

DOI: https://doi.org/10.7554/eLife.44821.034

*Figure 9 continued on next page*

*Figure 9 continued*

The following source data and figure supplements are available for figure 9:

**Source data 1.** Depletion of CD90.1+Foxp3- IL-10 competent Tr1 cells does not influence long-term tolerance to airway allergens.
DOI: https://doi.org/10.7554/eLife.44821.039
**Figure supplement 1.** Specificity and efficiency of using aCD90.1 for the depletion of IL-10 competent cells.
DOI: https://doi.org/10.7554/eLife.44821.035
**Figure supplement 1—source data 1.** Specificity and efficiency of using aCD90.1 for the depletion of IL-10 competent cells.
DOI: https://doi.org/10.7554/eLife.44821.036
**Figure supplement 2.** Characterization of CD3 negative CD90.1+ cell subsets.
DOI: https://doi.org/10.7554/eLife.44821.037
**Figure supplement 2—source data 1.** Characterization of CD3 negative CD90.1+ cell subsets.
DOI: https://doi.org/10.7554/eLife.44821.038

Together these data indicate that during primary allergen challenge CD90.1+ cells, the majority of which are Tr1-like cells but also include some CD90.1+Foxp3+, promote immune tolerance. However, endogenous Tr1-like cells do not contribute to tolerogenic memory in this model.

## Discussion

We have investigated the phenotypic stability and contributions to tolerogenic memory of endogenous Tr1-like cells in a mouse model of allergic airway inflammation-induced asthma. We report that natural Tr1-like cells only transiently express IL-10 after activation. Moreover, while cells that actively produce IL-10 are important for immune tolerance to airway allergens, neither depletion nor transfer of Tr1-like cells altered airway inflammation upon subsequent allergen challenge. Together these data suggest that naturally-arising Tr1-like cells may promote tolerance but do not contribute to a functionally stable tolerogenic memory population in this model.

Persistent antigenic signals were required for maintenance of IL-10 production and expression of *Irf1* and *Batf*, transcription factors previously linked to Tr1 status. It may be that repeated stimulation, a feature of many immunotherapy regimens, maintains active IL-10 production in airway Tr1-like cells by supporting expression of these canonical Tr1 transcription factors. In the gut, a prominent site of IL-10 production, antigen stimulation by commensal bacteria may provide a similar function (*Gagliani et al., 2013*; *Gagliani et al., 2015*; *Yu et al., 2017*).

In light of these data, we propose that IL-10 production is a temporary phenotype that a fraction of memory T cells adopt upon activation in the setting of certain previously defined differentiation signals (*Wu et al., 2011*; *Huang et al., 2017*; *Gregori et al., 2010*; *Li and Flavell, 2008*; *Coomes et al., 2017*; *Brockmann et al., 2017*). Furthermore, IL-10 production in response to allergen exposure may be stochastic and not necessarily predicated upon past production of IL-10. This model is perhaps consistent with past reports of phenotypic plasticity in induced Foxp3+ regulatory T cells (*Joetham et al., 2017*; *Hwang et al., 2018*), and between T-helper subsets in general (*Zhu and Paul, 2010*). Notably, the data presented here involving naturally arising Tr1-like cells stands in contrast to more phenotypically stable Tr1-like cells that have been engineered or differentiated in vitro (*Gregori and Roncarolo, 2018*).

Natural Tr1-like cells arise from the memory T-cell subset of mice previously sensitized to HDM. During the acute response, there is a significant increase in the frequency of CD90.1+Foxp3-, Tr1-like cells in the lungs and airways, the site of allergen challenge. After cessation of IL-10 production, these cells persisted over time in the lungs of challenged animals as tissue-resident memory cells. This is consistent with evidence that an immune reaction to an antigen can contribute to subsequent regulatory memory responses (*Brincks et al., 2013*; *Rosenblum et al., 2011*; *Sanchez et al., 2012*; *Sanchez Rodriguez et al., 2014*; *Gratz et al., 2013*).

Tr1-like cells in this model arose specifically in the lung, the site of allergen challenge and not in distal compartments such as the spleen. This is analogous to the report from Hondowicz et al. that tissue-resident Th2 cells drive allergic responses (*Hondowicz et al., 2016*). These data are also consistent with previous reports of tissue-specific roles for Tr1-like cells (though there are indications of geographic plasticity as well; *Yu et al., 2017*). However, currently the most commonly used routes for allergen-specific immunotherapy (SIT) are subcutaneous and oral. The data presented here

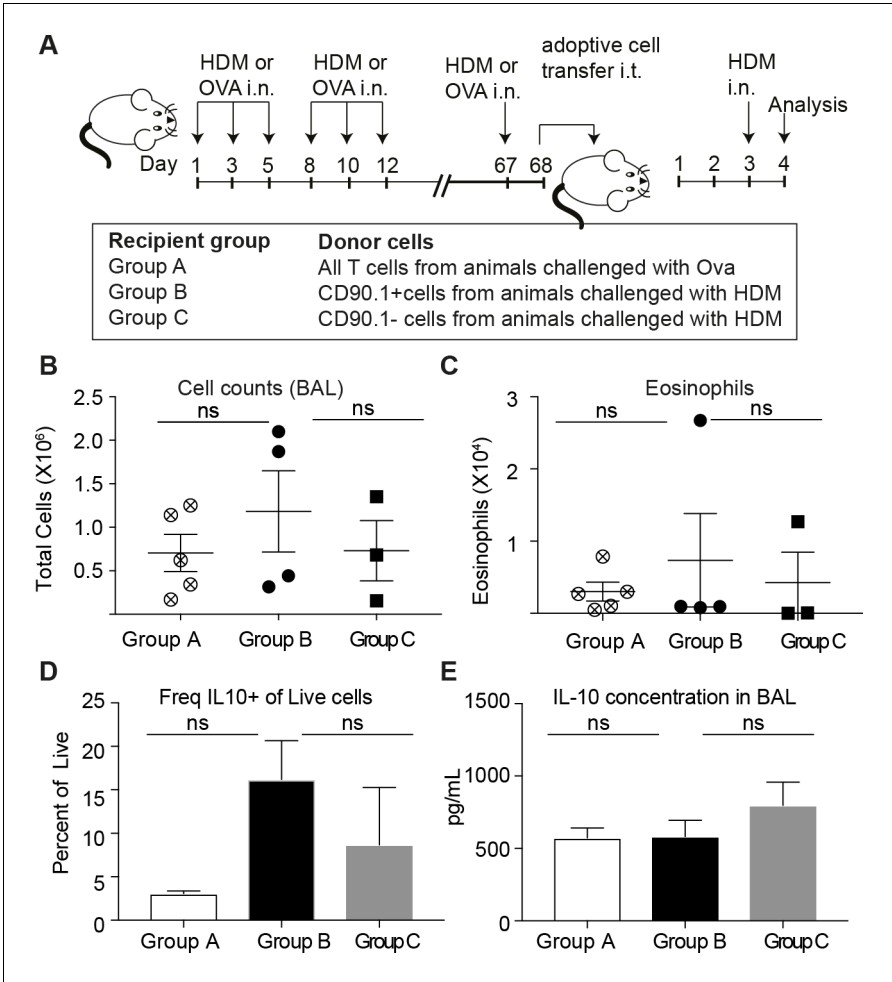

**Figure 10.** Transferred CD90.1+Foxp3- IL-10 competent Tr1 cells are not more likely than other T-cells to make IL-10 upon memory challenge to allergen. (**A**) All T cells or Tr1-like like cells or Effector T cells were isolated from lungs of HDM-sensitized animals after challenge and transferred into recipient groups as described. (**B**) Total cellular infiltration in the BAL and (**C**) eosinophilia in the BAL in recipients after HDM challenge is plotted. (**D**) Frequency of IL-10 expressing cells in lungs of recipient mice after 4 hr restimulation ex vivo with PMA/ionomycin/Brefeldin. (**E**) IL-10 ELISA from BAL fluid of recipient lungs. Error bars represent standard deviation of mean. n = 5 for group A, n = 4 For group B and n = 3 for group C.

DOI: https://doi.org/10.7554/eLife.44821.040

The following source data and figure supplements are available for figure 10:

**Source data 1.** Transferred CD90.1+Foxp3- IL-10 competent Tr1 cells are not more likely than other T-cells to make IL-10 upon memory challenge to allergen.
DOI: https://doi.org/10.7554/eLife.44821.045

**Figure supplement 1.** CD90.1 + CD4 T cells are functionally suppressive in vivo and in vitro.
DOI: https://doi.org/10.7554/eLife.44821.041

**Figure supplement 1—source data 1.** CD90.1 + CD4 T cells are functionally suppressive in vivo and in vitro.
DOI: https://doi.org/10.7554/eLife.44821.042

**Figure supplement 2.** Engraftment efficiencies in adoptive transfer studies.
DOI: https://doi.org/10.7554/eLife.44821.043

**Figure supplement 2—source data 1.** Engraftment efficiencies in adoptive transfer studies.
DOI: https://doi.org/10.7554/eLife.44821.044

suggest that intranasal routes may be more effective at inducing Tr1-like cells in the lung (*Takabayashi et al., 2003*).

Tr1-like cells and not Foxp3+ Treg were the major source of IL-10 in the lung in this model. Consistent with this, there was a greater increase in the frequency of Tr1-like cells versus Foxp3+ Tregs in the lung during the peak of inflammation. Unlike Tr1 cells, the majority of CD90.1-Foxp3+ Tregs in the inflamed lung were circulating, non-resident cells and their frequency did not increase upon memory allergen rechallenge.

Finally, Tr1-like cells and not Treg (irrespective of CD90.1+ expression) were also the predominant Derp1:I-A$^b$ tetramer-positive IL-10+ cell population in the lungs after secondary rechallenge. One could speculate that Foxp3+ regulatory memory Tregs may regulate inflammatory responses directed against self- rather than non-self-antigens, such as the HDM allergen (*Rosenblum et al., 2016*; *Rosenblum et al., 2011*). However, they may be specific for other HDM epitopes not assessed in our study. CD90.1+Foxp3+ cells are fewer in number and their frequency doesn't change significantly upon allergen sensitization. However, like the Tr1-like cells, they are also parenchymal and highly activated expressing CD44, Lag3, PD-1 and KLRG1.Furthermore they are also depleted when we use the CD90.1 antibody. Thus, while Tr1-like cells are the primary source of IL-10 in our model, we cannot rule out their secondary contribution to supressing inflammation in our acute model.

Tr1-like cells in our model expressed most of the cell surface markers previously associated with Tr1 cells including CD44, Lag3, PD-1, and KLRG1. However, these markers did not distinguish between Tr1-like cells and CD90.1+Foxp3+ Treg. Other studies have likewise reported that while these Tr1 markers identify a highly suppressive subset, they are not consistently expressed by all IL-10-producing Tr1-like cells (*Gagliani et al., 2013*; *Burton et al., 2014*; *White and Wraith, 2016*). This variation in their phenotypic markers is also consistent with the model that in vivo, natural Tr1-like cells may not be functionally stable in the absence of a persistent antigenic stimulus.

These findings may inform efforts to develop Tr1-based tolerogenic therapies. The phenotypic instability reported here may limit efforts to re-establish tolerance by promoting Tr1-based tolerogenic memory in the absence of repeated antigen stimulation. It may prove advantageous to engineer Tr1-like cells in vitro for adoptive transfer (*Gregori and Roncarolo, 2018*). Insight into the tissue-specific factors required for maintenance of Tr1 function in vivo will help better target both endogenous and engineered Tr1-like cells to induce long-term tolerance in patients.

# Materials and methods

## Key resources table

| Reagent type | Designation | Source of reference | Identifiers | Additional Information |
|---|---|---|---|---|
| Genetic Reagent (*M. musculus*) | *10BiT mice (Tg(Il10-Thy1)1Weav)* | PMID: 17694059 | RRID: MGI:3767675 | |
| Genetic Reagent (*M. musculus*) | *Foxp3eGFP (Foxp3tm2Tch)* | PMID: 15780990 | RRID: MGI:3699400 | |
| Genetic Reagent (*M. musculus*) | *B6.129(Cg)-Cd44$^{tm1Hbg}$/J* | PMID: 10528194 | RRID: MGI:4942279 | |
| Biological sample | Crushed House dust mite (Der p 1: 1911.78 mcg/vial, Endotoxin: 15900 UE/vial) obtained from *D. pteronyssinus* | Greer Laboratories Inc, Lenoir, NC | Cat: XPB70D3A25 Lot: 322781 | 20 ug house dust mite given intranasally |
| Biological sample | Crushed House dust mite (Der p 1: 2009.03 mcg/vial, Endotoxin: 9150 EU/vial) obtained from from *D. pteronyssinus* | Greer Laboratories Inc, Lenoir, NC | Cat: XPB70D3A25 Lot: 279019 | 20 ug house dust mite given intranasally |
| Antibody | Rat monoclonal CD4 (clone RM4-4), | Biolegend | Cat: 100552 | 1:100 for flow cytometry |

*Continued on next page*

*Continued*

| Reagent type | Designation | Source of reference | Identifiers | Additional Information |
|---|---|---|---|---|
| Antibody | Rat monoclonal CD3 (clone 17A2), | Biolegend | Cat: 100220 | 1:100 for flow cytometry |
| Antibody | Mouse monoclonal CD90.1 (Thy1.1) (clone OX-7) | Biolegend | Cat: 202516 | 1:150 for flow cytometry |
| Antibody | Rat monoclonal CD44 (clone IM7) | Biolegend | Cat: 103020 | 1:100 for flow cytometry |
| Antibody | Rat monoclonal Lag3 (clone C9B7W) | BD Biosciences | Cat: 562346 | 1:50 for flow cytometry |
| Antibody | Rat monoclonal CD49b (clone DX5) | eBiosciences | Cat: 12-5971-82 | 1:50 for flow cytometry |
| Antibody | Rat monoclonal CD25 (clone PC61) | Biolegend | Cat: 102026 | 1:100 for flow cytometry |
| Antibody | Syrian Hamster monoclonal KLRG1 (clone 2F1) | Biolegend | Cat: 138411 | 1:50 for flow cytometry |
| Antibody | Rat monoclonal PD1 (clone RMP1-30) | Biolegend | Cat: 109110 | 1:50 for flow cytometry |
| Antibody | Rat monoclonal IL-10 (clone JES5-16E3), | Biolegend | Cat: 505009 | 1:20 for flow cytometry |
| Antibody | Rat monoclonal IFNg (clone XMG1.2), | Biolegend | Cat: 505839 | 1:20 for flow cytometry |
| Antibody | Rat monoclonal IL-17 (clone TC11-18H10.1), | Biolegend | Cat: 506917 | 1:20 for flow cytometry |
| Antibody | Rat monoclonal IL-13 (clone eBio13A), | eBiosciences | Cat: 12-7133-41 | 1:20 for flow cytometry |
| Antibody | Rat monoclonal IL-4 (clone 11B11) | Biolegend | Cat: 504125 | 1:20 for flow cytometry |
| Antibody | Mouse monoclonal T-bet (clone 4B10) | Biolegend | Cat: 644813 | 1:20 for flow cytometry |
| Antibody | Rat monoclonal Gata-3 (clone TWAJ) | eBiosciences | Cat: 12-9966-42 | 1:20 for flow cytometry |
| Antibody | Mouse monoclonal CD90.1 (Thy1.1) (clone OX-7) | Biolegend | Cat: 202529 | 1:150 for flow cytometry |
| Antibody | Rat monoclonal CD11b (clone M1/70) | BD Biosciences | Cat: 562128 | 1:100 for flow cytometry |
| Antibody | Rat monoclonal MHC Class II (I-A/I-E) (clone M5/114.15.2) | eBiosciences | Cat: 86-5321-41 | 1:100 for flow cytometry |
| Antibody | Rat monoclonal Ly-6C (clone HK1.4) | Biolegend | Cat: 128041 | 1:200 for flow cytometry |
| Antibody | Mouse monoclonal CD64 (clone X54-5/7.1) | Biolegend | Cat: 139307 | 1:100 for flow cytometry |
| Antibody | Rat monoclonal Siglec F (clone S17007L) | Biolegend | Cat: 155505 | 1:200 for flow cytometry |
| Antibody | Rat monoclonal Ly-6G (clone 1A8) | Biolegend | Cat: 127617 | 1:200 for flow cytometry |
| Antibody | Armenian Hamster monoclonal CD11c (clone N418) | Biolegend | Cat: 117310 | 1:100 for flow cytometry |
| Antibody | Rat monoclonal CD24 (clone M1/69) | Biolegend | Cat: 101839 | 1:100 for flow cytometry |

*Continued on next page*

*Continued*

| Reagent type | Designation | Source of reference | Identifiers | Additional Information |
|---|---|---|---|---|
| Antibody | Rat monoclonal CCR7 (clone 4B12) | eBiosciences | Cat: 48-1971-82 | 1:100 for flow cytometry |
| Antibody | Rat monoclonal CD62L (clone MEL-14) | Biolegend | Cat: 104438 | 1:100 for flow cytometry |
| Antibody | Rat monoclonal CD44 (clone IM7) | Biolegend | Cat: 103032 | 1:100 for flow cytometry |
| Antibody | Rat monoclonal IL-7Ra (clone SB/199) | Biolegend | Cat: 121111 | 1:100 for flow cytometry |
| Antibody | Mouse monoclonal Thy1.1/CD90.1 (clone HIS51) | eBiosciences | Cat: 15-0900-82 | 1:150 for flow cytometry |
| Antibody | Armenian hamster monoclonal CD69 (clone H1.2F3) | Biolegend | Cat: 104511 | 1:100 for flow cytometry |
| Antibody | Mouse monoclonal KI-67 (clone B56) | BD Biosciences | Cat: 558615 | 1:100 for flow cytometry |
| Antibody | Armenian hamster monoclonal CD103 (clone 2E7) | Biolegend | Cat: 121431 | 1:100 for flow cytometry |
| Other | Derp1:I-Ab tetramer conjugated to PE | Gift from James moon | | 20 nM for flow |
| Antibody | Rat monoclonal CD45 AF700 clone 30-F11 (Biolegend, San Diego, CA) | Biolegend | Cat: 103128 | 3 ug/mouse |
| Antibody | InVivoMAb mouse monoclonal anti-mouse Thy1.1/CD90.1 (clone 19E12) | BioXcell | Cat: BE0214 | 200 ug/mouse |
| Chemical compound (drug) | Fingolimod (FTY720) | Sigma-Aldrich | Cat: SML0700-5MG | used at at 5 mg/kg, estimating 25 g per mouse |

## Mice

All animals used, including 10BiT/Foxp3eGFP, were on a C57BL/6 background. 10BiT/Foxp3eGFP mice were generated by crossing 10BiT mice (*Tg(Il10-Thy1)1Weav*) (**Maynard et al., 2007**) with Foxp3eGFP (*Foxp3tm2Tch*) (**Haribhai et al., 2007**) to homozygosity. The 10Bit strain contains a transgenic insertion consisting of the coding sequence of *Thy1a (Thy1.1, CD90.1)* followed by SV40 poly A sequence, inserted into a mouse Il10 gene contained in a BAC. The FOXP3EGFP strain co-express EGFP and the regulatory T cell-specific transcription factor *Foxp3* under the control of the endogenous promoter. CD44 knockout (*Cd44tm1Hbg*/J) animals were purchased from Jackson Laboratory. All mice were bred and maintained in house in a conventional facility according to institutional guidelines. All animal experiments and use procedures were approved by the Institutional Animal Care and Use Committee at Stanford University School of Medicine.

## Allergic airway inflammation model

Mice used for allergic airway sensitization ranged from 8 to 16 weeks of age. Mice were anesthetized with isoflurane (Henry-Schein, Dublin, OH) and challenged intranasally with 20 µg of house dust mite (HDM) extract (Greer Laboratories Inc, Lenoir, NC) in 50 µL sterile phosphate-buffered saline (PBS) on experimental days 1, 3, and five for sensitization and days 8, 10, and 12 for challenge. Endotoxin content of different lots varied but Derp1 content was the same. Only data from experiments using the same HDM lot were pooled. Single cell isolates from both the bronchoalveolar lavage (BAL) fluid and tissues were collected for staining and flow cytometry analysis of immune cell populations. For memory experiments, BAL fluid and tissues were collected 24 hr after animals were given an allergen rechallenge on day 67.

## Isolation of cells from tissues

Tissues were processed as described previously (*Gebe et al., 2017*). Briefly, bronchoalveolar lavages (BAL) were performed with 1 mL flushes of the lung with sterile PBS containing 0.2% BSA. BAL fluid was treated with ACK RBC lysis buffer (5 min, 37 degrees C) before cell counting and flow cytometry.

Following removal of BAL fluid, lungs were perfused with 5 mL sterile PBS injected into the right ventricle. Lungs were removed, placed in RPMI media and cut with scissors to approximately 1 mm pieces before the addition of collagenase IV (Sigma-Aldrich, St. Louis, MO) to 150 U/mL and DNase I (Sigma-Aldrich, St. Louis, MO) to 25 U/mL for digestion. Lung tissue was digested for 37 degrees C for 45 min on a shaker. A single cell suspension was obtained by pressing digested tissue through a 70 μm cell strainer using the plunger of a 3 mL syringe followed by a wash with PBS containing serum and EDTA. Where applicable, spleen and lymph nodes were isolated and separated into a single cell suspension using a 70 μm cell strainer similarly. All tissues were treated with ACK RBC lysis following single cell suspension (5 min, 37 degrees C) for cell counting and flow cytometry.

## Histology

Histology was performed as described previously (*Yadava et al., 2016*). Lungs were inflated with 700–800 μl of 10% neutral buffered formalin, embedded into paraffin. 5 μM thick sections were stained with haematoxylin (Sigma-Aldrich, St. Louis, MO) and eosin (Merck Millipore, Burlington, MA). Stained slides were imaged by light microscopy.

## Flow cytometry and antibodies

Cell subsets were distinguished by surface staining for CD4 (clone RM4-4), CD3 (clone 17A2), CD90.1 (Thy1.1) (clone OX-7) and Foxp3 eGFP. Surface expression of canonical Tr1 markers was assessed by surface staining for CD44 (clone IM7), Lag3 (clone C9B7W), CD49b (clone DX5), CD25 (clone PC61), KLRG1 (clone 2F1), and PD1 (clone RMP1-30). All antibodies were purchased from Biolegend (San Diego, CA).

To analyse cytokine production, cells from lung digests were stimulated with 20 ng/mL PMA (Sigma-Aldrich, St. Louis, MO), 1 μg/mL Ionomycin (Sigma-Aldrich, St. Louis, MO) and 3 μg/mL Brefeldin A (eBioscience-ThermoFisher, Waltham, MA) for 4 hr at 37 degrees C. Cells were surface stained and then were fixed with 2% PFA. Cells were then stained intracellularly with IL-10 (clone JES5-16E3), IFNg (clone XMG1.2), IL-17 (clone TC11-18H10.1), IL-13 (clone eBio13A), IL-4 (clone 11B11), T-bet (clone eBio4B10), and Gata-3 (clone TWAJ) in permeabilization buffer (eBioscience-ThermoFisher, Waltham, MA). Derp1:I-A[b] tetramer conjugated to PE was a kind gift from James Moon (*Hondowicz et al., 2016*). Cells were stained with 20 nM tetramer for 1 hr at room temperature prior to surface staining. Stained cells were acquired on BD FACS LSRII instruments in the Stanford Shared FACS Facility and analyzed using FlowJo software (FlowJo, LLC, Ashland, OR).

## Intravascular labeling of cells

For staining of non-tissue-resident circulating cells in the lung, CD45 AF700 clone 30-F11 (Biolegend, San Diego, CA) was injected retro-orbitally 2 min before mice were euthanized (*Anderson et al., 2014*).

## Blocking lymphocyte migration using FTY720

To block lymphocyte egress from lymph nodes, HDM-sensitized animals were given daily IP injections of FTY720 (Sigma-Aldrich Cat: SML0700-5MG) at 5 mg/kg, estimating 25 g per mouse, or DMSO (sham control) in saline from day 64 to day 68 (*Brinkmann et al., 2002*).

## Depletion of IL-10-producing cells

Depletion of IL-10-producing cells was performed by intraperitoneal injection of 200 ug of anti-Thy1.1 antibody or matched isotype control (BioXCell, West Lebanon, NH). This was done on day eight for the acute model (as depicted in the schematic in *Figure 8A*) and beginning on day 64 in the memory model (as depicted in the schematic in *Figure 8F*).

## In vitro Tr1 cultures for cell sorting and cytokine capture assays and qPCR

Cells were isolated from spleens and lymph nodes of two mice, as described previously (*Gebe et al., 2017*). Naïve CD4 T cells were isolated using the EasySep Mouse Naïve CD4+ T Cell Isolation Kit (STEMCELL, Cambridge, MA) following manufacturer's instructions. Tr1 cells were differentiated in vitro as described previously (*Chihara et al., 2016*). Briefly, naïve CD4 T cells were cultured in Tr1 clone medium with murine IL-27 (25 ng/mL) on plates coated with anti-CD3 (2 ug/mL) and anti-CD28 (2 ug/mL) for 3 days. Cells were then re-activated on fresh anti-CD3/anti-CD28-coated plates in media without IL-27 for 4.5 hr to promote IL-10 secretion. IL-10 secreting cells were labeled using the Mouse IL-10 Secretion Assay Detection Kit (Miltenyi, Auburn, CA) following manufacturer's instructions and surface stained for CD3, CD4, CD90.1, and viability for sorting.

Cells were sorted on the BD FACS Aria II instruments in the Stanford Shared FACS Facility and processed using the TaqMan Gene Expression Cells-to-Ct kit (ThermoFisher, Waltham, MA), following manufacturer's instructions, to generate cDNA for analysis. Gene expression was quantified using PrimeTime qPCR Probe Assays from IDT (Coralville, IA) for Irf1 (Mm.PT.58.33516776), Batf (Mm.PT.58.33231426), and beta-Actin (Mm.PT.39a.22214843.g) and SensiFAST Probe Hi-ROX Kit (Bioline, Memphis, TN) for 40 cycles. Gene expression was calculated as a delta Ct value with beta-Actin as a reference gene and displayed as fold-change in expression normalized to the CD90.1-IL-10- sort group.

## Adoptive transfer studies

10BiT/Foxp3eGFP mice were sensitized with either HDM (n = 20) or ovalbumin (n = 5) following the standard memory protocol and rechallenged at day 67 post initial challenge with their respective antigen. Lungs were digested and pooled, and CD4+ T cells were isolated using the EasySep Mouse CD4+ T cell isolation kit (STEMCELL, Cambridge, MA). CD4+ T cells from ovalbumin-sensitized mice served as an antigen-inexperienced control (Group A). CD4+ T cells from HDM-sensitized mice were then separated using the EasySep Mouse CD90.1 Positive Selection Kit into CD90.1-enriched (Tr1, Group B) and CD90.1-depleted (effector, Group C) donor populations. These cells were transferred into naïve C57Bl/6 mice intratracheally. Group A mice received 1.53E6 antigen-inexperience CD4 T cells per mouse, Group B mice received 1.36E4 Tr1 enriched CD4 T cells, and Group C mice received 2.51E5 Tr1-depleted CD4 T cells. All groups were challenged with HDM 48 hr after transfer and BAL and lungs were analyzed.

## Statistics

All graph generation and statistical analyses were performed using GraphPad Prism (GraphPad Software, Inc, La Jolla, CA). Statistical tests used are specified in each figure legend. $p < 0.05$ was considered statistically significant.

## Acknowledgements

This work was supported in part by National Institutes of Health grants R01 DK096087-01, R01 HL113294-01A1, and U01 AI101984 to PLB. Koshika Yadava was supported by the Swiss National Science Foundation early postdoctoral mobility grant and Child Health Research Institute and the Stanford NIH-NCATS-CTSA (grant no. UL1 TR001085). Carlos Medina was supported by the Stanford Molecular and Cellular Immunobiology Training Grant (5 T32 AI07920, PI Olivia Martinez), the Ford Foundation Pre-Doctoral Fellowship and the Stanford Diversifying Academia, Recruiting Excellence Fellowship. We thank Maria-Grazia Roncarolo, Rosa Bacchetta, Gernot Kaber and Graham Ogg for their helpful comments. The authors declare no competing financial interests

## Additional information

### Funding

| Funder | Grant reference number | Author |
| --- | --- | --- |
| National Institutes of Health | R01 DK096087-01 | Paul L Bollyky |

| National Institutes of Health | R01 HL113294-01A1 | Paul L Bollyky |
|---|---|---|
| National Institutes of Health | U01 AI101984 | Paul L Bollyky |
| Swiss National Science Foundation | Early postdoctoral mobility grant | Koshika Yadava |
| Child Health Research Institute Stanford | UL1 TR001085 | Koshika Yadava |
| Ford Foundation | | Carlos Obed Medina |
| Stanford University | Molecular and Cellular Immunobiology Training Grant 5 T32 AI07920 | Carlos Obed Medina |
| Stanford University | Diversifying Academia, Recruiting Excellence | Carlos Obed Medina |

The funders had no role in study design, data collection and interpretation, or the decision to submit the work for publication.

## Author contributions

Koshika Yadava, Conceptualization, Data curation, Formal analysis, Supervision, Funding acquisition, Validation, Investigation, Methodology, Writing—original draft, Writing—review and editing; Carlos Obed Medina, Conceptualization, Data curation, Formal analysis, Funding acquisition, Validation, Methodology, Writing—original draft, Writing—review and editing; Heather Ishak, Irina Gurevich, Ievgen O Koliesnik, Investigation, Methodology; Hedwich Kuipers, Investigation, Writing—review and editing; Elya Ali Shamskhou, Investigation; James J Moon, Casey Weaver, Methodology, Writing—review and editing; Kari Christine Nadeau, Supervision, Investigation, Project administration, Writing—review and editing; Paul L Bollyky, Conceptualization, Formal analysis, Supervision, Funding acquisition, Investigation, Writing—original draft, Project administration, Writing—review and editing

## Author ORCIDs

Koshika Yadava https://orcid.org/0000-0002-5827-9177
Carlos Obed Medina https://orcid.org/0000-0002-8324-7994
James J Moon https://orcid.org/0000-0001-8246-931X
Paul L Bollyky https://orcid.org/0000-0003-2499-9448

## Ethics

Animal experimentation: All animal experiments and use procedures were approved by the Institutional Animal Care & Use Committee at Stanford University School of Medicine (APLAC 27657).

## Decision letter and Author response

Decision letter https://doi.org/10.7554/eLife.44821.048
Author response https://doi.org/10.7554/eLife.44821.049

# Additional files

## Supplementary files

• Transparent reporting form
DOI: https://doi.org/10.7554/eLife.44821.046

## Data availability

All data generated or analysed during this study are included in the manuscript and supporting files.

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
