## [Decision Letter]

[Editors’ note: this article was originally rejected after discussions between the reviewers, but the authors were invited to resubmit after an appeal against the decision.]

Thank you for submitting your work entitled "Natural Tr1-like cells do not confer long-term tolerogenic memory" for consideration by *eLife*. Your article has been reviewed by three peer reviewers, one of whom is a member of our Board of Reviewing Editors, and the evaluation has been overseen by a Senior Editor. The following individuals involved in review of your submission have agreed to reveal their identity: Benoit Salomon (Reviewer #1).

Our decision has been reached after consultation between the reviewers. Based on these discussions and the individual reviews below, we regret to inform you that your work will not be considered further for publication in *eLife*.

*Reviewer #1:*

In this manuscript, Yadava et al., studied the role of Tr1-like cells (defined by the Foxp3- CD90.1/IL-10+ phenotype using 10BiT BAC transgenic mice to identify cells that produced or had produced IL-10) in a mouse model of lung allergy to the house dust mite allergen. They found that, at the peak of the disease (day 16), Tr1-like cells are the main IL-10 producer cells (Figure 1F,G and Figure 2B). However, the other main findings (role of antigen persistence to maintain IL-10 expression by Tr1-like cells (Figure 4), role of Tr1-like cells assessed in depleting experiment (Figure 8), role of Tr1-like cells assessed in transfer experiment (Figure 9) are over-interpreted for the reasons explained below.

Essential revisions:

1) The most critical experiments of this manuscript have been done only once with 3-5 mice per group (Figure 4, Figure 8 and Figure 9, and probably (not indicated) also Figure 2 and Figure 5) and all the other experiments have been only two times. Whereas two times is borderline but acceptable when results are striking, which is the case here for some experiments, performing an experiment only one time is unacceptable. That is why experiments showing the existence of "former" Tr1-like cells or showing that Tr1-like cells are critical at the peak of the disease but no more during memory are over-interpreted.

2) Another major point challenges the conclusions of the existence of "former" Tr1-like cells (Figure 4). The "former" Tr1 like cells, which are CD90.1+ that no more produce IL-10, represent {plus minus}75% at day 16 (peak of the disease) and 5-10% at day 30. The authors concluded from this finding that Tr1-like cells lost their capacity to produce IL-10 because of lack of Ag-stimulation. Another interpretation of the data is preferential survival or proliferation of IL-10- CD90.1+ cells over IL-10+ CD90.1+ cells. in vitro experiment depicted in Figure 5 does not help to solve this issue because it is a totally different system and we only have proportions and not absolute numbers that may help to distinguish between these two mechanisms (IL-10 loss vs preferential survival or proliferation).

3) Another major point challenges the conclusions that Tr1-like cells are dispensable to maintain immune tolerance, shown in the depleting experiment (Figure 8). Indeed, it is very likely that the "former" Tr1-like cells loose CD90.1 expression after silencing of the IL-10 promoter over time. Thus, depleting CD90.1+ cells would not deplete these "former" Tr1-like cells. To address this major concern, critical data are missing. What is the half-life of CD90.1 (both mRNA and protein) when the gene is no more transcripted. Also, we do not know how profound the depletion of CD90.1+ cells is, since the authors have not checked whether the depleting mAb could mask the CD90.1 staining.

4) Another major point challenges the conclusions that Tr1-like cells are dispensable to maintain immune tolerance, shown in the transfer experiment (Figure 9). The positive control is missing here. The authors should show that transferring Tr1-like cells purified from day 16 (peak of the disease) are able to control the disease. Without this positive control, we do not know if the cell transfer experiment is working. Also, it would be better to have a normal model of lung allergy by injecting the allergen 6 times (and not 4 times that give a weak disease).

5) The term "resident" for CD45- cells in the blood partitioning experiment is non-appropriate. The term "parenchymal" is more accurate since we do not if they are resident, before the experiment injecting FTY720. Also, we do not know how efficient, was the injection of FTY720. Critical control data, for instance looking at T cell numbers in the blood, are lacking. Data of Figure 7D to support FTY720 efficacy are too weak.

6) The data on Irf1 and Batf are very weak. Only one experiment? No statistics.

*Reviewer #2:*

IL-10-producing Tr1 cells have gained much interest because of their activity to suppress inflammatory responses, but it remains unclear whether they are functionally stable and capable of mediating tolerogenic memory. In the present study, Yadava et al. addressed this important question using a HDM model of allergic lung inflammation. They took advantage of IL-10-Thy1.1 BAC Tg x Foxp3-eGFP reporter mice to identify IL-10-expressing CD4^+^ T cell subsets in primary and secondary (recall) responses to HDM and to examine whether Tr1 cells play any role in the primary and secondary responses. The main results are; (1) during the primary response, Thy1.1+Foxp3-CD4^+^ T cells accumulate within the lung tissue but decrease in numbers and stop producing IL-10 after the peak of inflammation; (2) in the recall response, antigen-specific "memory" Thy1.1+Foxp3-CD4^+^ T cells can be detected within the lung tissues even in FTY720-treated mice; (3) while depletion of Thy1.1+ cells exacerbated inflammation in the primary response, their depletion failed to affect inflammation in the recall response; (4) adoptive transfer of "memory" Thy1.1+Foxp3-CD4^+^ T cells to naïve mice resulted in loss of Thy1.1 expression upon re-challenge and failed to affect inflammation. Based on these results, the authors conclude that Tr1 cells are not functionally stable and do not contribute to "tolerogenic memory" in this experimental setting.

In my view, there is one fundamental flaw that seriously questions the validity of the conclusions of this study. Thus, the authors' definition of Tr1 cells as being Thy1.1+Foxp3-CD4^+^ T cells is inappropriate. Because IL-10 can be produced by different CD4^+^ T cell subsets, particularly Th2 cells, it is absolutely required to include other Tr1 cell markers including expression of CD49b and LAG3 as well and to exclude Th2 cells by using an IL-4 reporter to more rigorously define Tr1 cells. This is particularly important because the authors used a model of Th2 inflammation. It is also absolutely necessary to demonstrate that their "Tr1" or "Tr1-like" cells show suppressive activities. The exacerbated primary response to HDM after depletion of Thy1.1+ cells could be accounted for solely by depletion of Thy1.1+Foxp3+ Treg cells. No data demonstrate that Tr1 cells are actually generated in meaningful numbers in this experimental system. If so, the use of this experimental system is certainly inappropriate to address the main question of this study.

Essential revisions:

1) Figure 6, Figure 7, Figure 8 and Figure 9: The authors should characterize the dynamics of CD4^+^ T cell subsets and of inflammatory responses during the recall response. It is surprising that the authors have analyzed mice only 1 day after the recall immunization. It is possible that depletion or transfer of Thy1.1+Foxp3-CD4^+^ T cells may affect inflammation when examined at later time points.

2) Figure 6: A group of animals that have been primed by HDM but are not given HDM on day 67 is necessary. In the absence of this control, it is impossible to judge whether the observed responses in the experimental group reflect the recall response or the remnant of the primary response.

3) Figure 7: Because FTY720 inhibits egress of lymphocytes from secondary lymphoid tissues to blood circulation, it is possible that Thy1.1+Foxp3- T cells do not originate from tissue resident memory cells but are recruited from effector memory cells that are circulating in the blood. To claim that they are derived from tissue-resident memory cells, parabiosis experiments are necessary.

4) Figure 6G, Figure 7—figure supplement 1: To validate the specificity of Derp1 tetramer staining, staining with a control tetramer (i.e., a tetramer loaded with an irrelevant peptide) should also be performed.

5) Figure 8: It is not clear whether depletion of Thy1.1+ cells was properly assessed. It seems that anti-Thy1.1 (clone OX7) was used for flow cytometric analyses of T cells from anti-Thy1.1 (or isotype control) mAb injected animals. Because the anti-Thy1.1 mAb used for depletion should compete with the anti-Thy1.1 mAb used for flow cytometric analyses, the apparent absence of "Thy1.1+" cells in the treated group may not be due to their depletion but could simply result from the competition of the two clones of antibodies. This possibility should be ruled out.

6) Figure 8C,D: The magnitude of the response of the control (yet HDM-immunized) group is much less than that of the HDM-immunized group shown in Figure 1B, C but rather similar to the non-immunized group. There must be something wrong.

*Reviewer #3:*

The authors demonstrate the phenotype of lung resident Tr1-like cells during mite sensitization. They clearly show the stability and the origin of lung Tr1 cells using an in vitro assay system. However, some issues need to be further addressed to confirm and strengthen their claims on the role of Tr1 cells in the lung.

Essential Revisions:

1) The authors claim that the major source of lung IL-10 is Tr1-like cell after mite immunization because a half of IL10 positive cells is Foxp3-CD4^+^ (Figure 3B). However, total number of cells is also important to assess the contribution of IL-10 production. The total number of each cell population which produces IL-10 should be shown. The amount of IL-10 expression from each cell should also be addressed. In Figure 1, signal intensity of Thy1.1 (IL10) of Treg is higher than that of Tr1-like cells whereas the percentage is lower. As the original paper of the IL-10 reporter mice indicated that MFI of Thy1.1 can be used to estimate the IL-10 producing potential, they should summarize the MFI of Thy1.1 as the capacity of IL-10 expression, in addition to the percentage, of each cell population in order to support possible contribution of Tr1-like cells to the amount of lung IL-10.

2) They appear to have carried out the in vivo labeling experiment without immunization in Figure 2 because IL-10+ cells are not so high (Figure 2C). As the origin of Tr1 cells during inflammation is more important, they have to address the issue under immunization.

3) In Figure 5, they compare IL-10 production with or without in vitro TCR stimulation. Is it possible to maintain primary T cells in vitro without stimulation for 5 days? They should show the percentage of living cells after culture. If unstimulated cells are not normal, it's difficult to conclude that IL-10 production is dependent on TCR stimulation.

4) In Figure 4G, they examine cytokine production from Thy1.1+ cells. Because IL-4 is a major Th2 cytokine, they should assess the production of IL-4.

5) In Figure 7, they use FTY720 for inhibiting the egress of lymphocyte. As they only show the ratio of naïve/effector fraction in Figure 5F, it is difficult to understand the dynamics of CD4^+^ subsets. They should summarize the total number of each T cell subset with or without FTY720 treatment.

6 In Thy1.1 depletion experiments (Figure 8), not only Tr1 but also other IL-10 producing cell (e.g.; DCs, B cells and macrophages) population would be deleted. They should address the effects on other IL-10 producing cells, at least show the percentage and the number of other cell populations after anti-Thy1.1 treatment.

[Editors’ note: what now follows is the decision letter after the authors submitted for further consideration.]

Thank you for resubmitting your work entitled "Natural Tr1-like cells do not confer long-term tolerogenic memory" for further consideration at *eLife*. Your revised article has been favorably evaluated by Tadatsugu Taniguchi (Senior Editor) and the Reviewing Editor.

The manuscript has been improved but there are some remaining issues that need to be addressed before acceptance, as outlined below:

The authors have properly responded to the reviewers' comments. One concern is their interpretation of the results in the depletion experiments in Figure 9 and Figure10, which are related to whether IL-10-producing T cells indeed bear in vivo suppressive activity in the allergy model. This issue was raised by one of the reviewers, but the authors' response is not sufficiently convincing. In Figure 9A, B, there is still a possibility that depletion of Thy-1.1+Foxp3+ cells by anti-Thy-1.1 antibody is responsible for the exacerbation of the allergic response because these Foxp3+ cells are KLRG1+ (i.e., highly activated) even if the cell number is small as shown in Figure 4. Suppressive activity shown with in vitro induced Tr1 cells is not a proper support for their claim. However, the issue does not oppose their main claim that Tr-1-like cells do not possess suppressive activity on recall responses. Modification of their interpretation may be sufficient for final acceptance.

---

## [Author Response]

[Editors’ note: the author responses to the first round of peer review follow.]

Reviewer #1:[…] Essential revisions:1) The most critical experiments of this manuscript have been done only once with 3-5 mice per group (Figure 4, Figure 8 and Figure 9, and probably (not indicated) also Figure 2 and Figure 5) and all the other experiments have been only two times. Whereas two times is borderline but acceptable when results are striking, which is the case here for some experiments, performing an experiment only one time is unacceptable. That is why experiments showing the existence of "former" Tr1-like cells or showing that Tr1-like cells are critical at the peak of the disease but no more during memory are over-interpreted.

Thank you for bringing this to our attention. However, this appears to be a misunderstanding. The experiments in question were instances where we did not list the number of experimental replicates. We have now included information on the number of experimental replicates throughout the manuscript.

2) Another major point challenges the conclusions of the existence of "former" Tr1-like cells (Figure 4). The "former" Tr1 like cells, which are CD90.1+ that no more produce IL-10, represent {plus minus}75% at day 16 (peak of the disease) and 5-10% at day 30. The authors concluded from this finding that Tr1-like cells lost their capacity to produce IL-10 because of lack of Ag-stimulation. Another interpretation of the data is preferential survival or proliferation of IL-10- CD90.1+ cells over IL-10+ CD90.1+ cells. in vitro experiment depicted in Figure 5 does not help to solve this issue because it is a totally different system and we only have proportions and not absolute numbers that may help to distinguish between these two mechanisms (IL-10 loss vs preferential survival or proliferation).

Our conclusion that former Tr1 cells exist is based on data from 10Bit mice, a well characterized IL-10 reporter stain used in many studies in the past 12 years (Maynard and Weaver, 2008; Maseda et al., 2012; Moreira-Teixeira et al., 2017; Bouabe; Clement et al., 2016). In this reporter strain, CD90.1 is expressed only after IL-10 production. The presence of CD90.1 on Foxp3- Tcells in the absence of active IL-10 production can therefore only mean that these cells previously produced this cytokine but they do not currently – that is to say that they are former Tr1.

Regarding the possibility that these former TR1 cells are over represented in our model due to ongoing proliferation, it is difficult to see how this could be the case in the absence of cognate antigen for a month and given the overall marked decline in the airway inflammatory cells during this time (Figure 5B). If the reviewer is instead suggesting that these CD90.1+ cells represent de novo proliferative responses of naïve cells to different (perhaps self?) antigens, this seems highly unlikely given that a large fraction of former Tr1 are memory T cells (Figure 5I). Nonetheless, to directly interrogate whether these former Tr1 show evidence of extensive proliferation, we examined Ki67 expression and found that the large majority (>90%) were Ki67 negative (Figure 8—figure supplement 2). These data do not support extensive proliferation in former Tr1.

Regarding the possibility that Tr1 cells survive poorly in post-inflammatory tissues, we agree that this is possible. However, we do not see how it would challenge the conclusion that IL-10CD90.1+ T-cells are former Tr1. Moreover, this would fit well with our model because antigenic signals govern both T cell survival (Freitas and Rocha, 1999) as well as IL-10 production (Roncarolo et al., 2006). Nonetheless, to test this directly we examined the survival of purified IL10+CD90.1+ tr1 cells and find that ongoing antigenic signals are critical for Tr1 survival. These data are now included as Figure 6—figure supplement 1.

3) Another major point challenges the conclusions that Tr1-like cells are dispensable to maintain immune tolerance, shown in the depleting experiment (Figure 8). Indeed, it is very likely that the "former" Tr1-like cells loose CD90.1 expression after silencing of the IL-10 promoter over time. Thus, depleting CD90.1+ cells would not deplete these "former" Tr1-like cells. To address this major concern, critical data are missing. What is the half-life of CD90.1 (both mRNA and protein) when the gene is no more transcripted.

The kinetics of CD90.1 expression and IL-10 in the 10Bit model were previously addressed in many studies (Maynard et al., 2007; Moreira-Teixeira et al., 2017; Maseda et al., 2012). In particular, it was reported that CD90.1 mRNA and IL-10 mRNA follow parallel dynamics after stimulation (Maseda et al., 2012) and that cell surface CD90.1 expression can persist for months (Maseda et al., 2012; Maynard et al., 2007). In our own experiments, we find that on Day 68 of our rechallenge protocol 3.7% of CD4^+^ T-cells in mice originally treated with HDM are CD90.1+ (Figure 7E). The CD90.1+Foxp3- subset memory cells at this time are predominantly memory cells (Figure 8—figure supplement 2). These are the cells that our depletion regimen could be expected to deplete.

Also, we do not know how profound the depletion of CD90.1+ cells is, since the authors have not checked whether the depleting mAb could mask the CD90.1 staining.

To address this, we have used a distinct clone (Ox-7) to stain cells as compared to the antibody clone (19E12) which was used for depletion. We have included these data in figure 9—figure supplement 1, which shows that the depletion antibody does not mask the staining epitope of Ox7 clone as well as another fluorescently tagged CD90.1 clone (HIS51). To further support the efficient depletion of these cells in our system, we have also included data that show intracellular IL-10 protein is reduced in depleted groups (Figure 9—figure supplement 1)

4) Another major point challenges the conclusions that Tr1-like cells are dispensable to maintain immune tolerance, shown in the transfer experiment (Figure 9). The positive control is missing here. The authors should show that transferring Tr1-like cells purified from day 16 (peak of the disease) are able to control the disease. Without this positive control, we do not know if the cell transfer experiment is working.

To address this and to provide a positive control for these experiments, we adoptively transferred 1 million in vitro induced Tr1 cells intratracheally into HDM sensitized recipients. We observed a decrease in total BAL counts and airway eosinophilia. We also showed that Tr1 cells are suppressive in vitro. (Figure 10—figure supplement 1).

Also, it would be better to have a normal model of lung allergy by injecting the allergen 6 times (and not 4 times that give a weak disease).

This appears to be a misunderstanding. All of our experiments involved exposure to allergen 6 times. This point has been made clear in the methods section and in our graphics.

5) The term "resident" for CD45- cells in the blood partitioning experiment is non-appropriate. The term "parenchymal" is more accurate since we do not if they are resident, before the experiment injecting FTY720.

Fingolimod (FTY720) has been widely used to investigate the role of tissue-resident cells in disease models (Hofmann et al., 2006; Janssen et al., 2015), including in the lungs (Connor et al., 2010). In the large majority of these studies, the term ‘resident’ is used for the cells spared by FTY720 treatment. We nonetheless appreciate the distinction between resident versus parenchymal cells is important and we now refer to these cells as parenchymal.

Also, we do not know how efficient, was the injection of FTY720. Critical control data, for instance looking at T cell numbers in the blood, are lacking. Data of Figure 7D to support FTY720 efficacy are too weak.

To examine the effectiveness of FTY720 in depleting circulating cells in this model, we have included data on the absolute numbers of CD62L+ cells in mice treated with saline control vs. FTY720 as well as the absolute numbers of Tr1 and T conventional cells in different tissues (Figure 8—figure supplement 1). We find significantly fewer CD62L+ T cells in lungs of FTY720 mice compared to controls. In contrast there was no difference in the draining lymph nodes and spleen. We are confident that these data now strongly support the conclusion that FTY720 depletes circulating cells in this model as it has in other models (Hofmann et al., 2006; Connor et al., 2010; Janssen et al., 2015).

6) The data on Irf1 and Batf are very weak. Only one experiment? No statistics.

Each data point in the bar graph in question actually represents a separate experiment (mean relative expression in triplicate) and the data are pooled from five independent experiments. We have now included information on the number of experiments and replicates in the figure legend as well as statistical analysis.

Reviewer #2:[…] In my view, there is one fundamental flaw that seriously questions the validity of the conclusions of this study. Thus, the authors' definition of Tr1 cells as being Thy1.1+Foxp3-CD4^+^ T cells is inappropriate. Because IL-10 can be produced by different CD4^+^ T cell subsets, particularly Th2 cells, it is absolutely required to include other Tr1 cell markers including expression of CD49b and LAG3 as well and to exclude Th2 cells by using an IL-4 reporter to more rigorously define Tr1 cells. This is particularly important because the authors used a model of Th2 inflammation.

To determine whether the IL-10 producing cells in our model are Tr1 or Th2 we examined the expression of multiple Th2 markers including IL-4, IL-13, and GATA3. We find that none of these are present on our putative Tr1 cells (Figure 7—figure supplement 2). These data conclusively demonstrate that our Tr1-like cells are not Th2. Furthermore, we demonstrate that the Tr1 cells in this system are suppressive (Figure 10—figure supplement 1). As Th2 cells would not be expected to be suppressive, this further establishes that we are dealing with Tr1. Given the absence of any other Th2 markers, the suppressive capabilities of our Tr1, and the lack of any IL4 co-staining with our Tr1 in this system, we do not see the benefit of generating a triple reporter mouse with the capacity to track IL-4 as well as IL-10 and Foxp3.

Regarding the CD49b and LAG3 markers, in our model we find that CD49 is not associated with IL-10 expression while LAG3 is upregulated on all IL-10 producing cells including Foxp3+ Treg (Figure 4). We conclude that CD49 and LAG3 do not characterize IL-10+Foxp3- T-cells in this model and that the use of CD49b and LAG3 as Tr1 markers in this system is therefore not justified. We appreciate that this result contradicts data from some other models, notably the IL-10-GFP reporter strain. However, the reviewer may be aware that there is an active debate regarding the defining characteristics of Tr1 cells. In particular, CD49b and LAG3 have been identified as specific Tr1 markers in some studies and not in others (Huang et al., 2018; Burton et al., 2014). We hope the reviewer agrees that data should always be considered on its own merits and that input from different models and perspectives ultimately benefits the field.

It is also absolutely necessary to demonstrate that their "Tr1" or "Tr1-like" cells show suppressive activities. The exacerbated primary response to HDM after depletion of Thy1.1+ cells could be accounted for solely by depletion of Thy1.1+Foxp3+ Treg cells. No data demonstrate that Tr1 cells are actually generated in meaningful numbers in this experimental system. If so, the use of this experimental system is certainly inappropriate to address the main question of this study.

To address this point and to provide a positive control for these experiments, we adoptively transferred 1 million in vitro induced Tr1 cells intratracheally into HDM sensitized recipients. We observed a decrease in total BAL counts and eosinophil counts. We also show that the Tr1 cells are also suppressive in vitro. These data are included in Figure 10—figure supplement 1.

Essential revisions:1) Figure 6, Figure 7, Figure 8 and Figure 9: The authors should characterize the dynamics of CD4^+^ T cell subsets and of inflammatory responses during the recall response. It is surprising that the authors have analyzed mice only 1 day after the recall immunization. It is possible that depletion or transfer of Thy1.1+Foxp3-CD4^+^ T cells may affect inflammation when examined at later time points.

We examined recall responses after 1 day because this is the endpoint most relevant to clinical allergic asthma and to the studies of Tr1 cells planned or underway in human clinical studies. Furthermore, while fibrosis and airway remodeling can occur in chronic human asthma, the relevance of the available mouse models (or of Tr1 cells) to this pathophysiology is unclear (Moore and Hogaboam, 2008). Given these considerations, we did not feel that the proposed experiments were likely to make a productive contribution to the present work.

2) Figure 6: A group of animals that have been primed by HDM but are not given HDM on day 67 is necessary. In the absence of this control, it is impossible to judge whether the observed responses in the experimental group reflect the recall response or the remnant of the primary response.

Given the normalization of BAL counts and Eosinophilia we observed by day 30 (Figure 4B) it was unclear to us why the reviewer believes that airway inflammation would not also have normalized on day 67.

Nonetheless, to address this, we sensitized mice with HDM or PBS control and waited until day 68 before analyzing BAL and lung tissues without rechallenge. We find no difference in total BAL counts (Figure 8—figure supplement 2). This demonstrates that the airway inflammatory response seen on day 68 following rechallenge is not a remnant of primary response but rather a rapid memory response to allergen re-exposure.

3) Figure 7: Because FTY720 inhibits egress of lymphocytes from secondary lymphoid tissues to blood circulation, it is possible that Thy1.1+Foxp3- T cells do not originate from tissue resident memory cells but are recruited from effector memory cells that are circulating in the blood. To claim that they are derived from tissue-resident memory cells, parabiosis experiments are necessary.

While is true that FTY720 treatment prevents lymphocyte egress from secondary lymphoid organs, it is perhaps less well known that this drug actually targets multiple S1P s and has broader effects on leukocyte migration. In particular, it also targets endothelial cell barrier function and prevents entry of cells into lungs, CNS and other tissues (Hawksworth et al., 2012; Hutchison et al., 2009; Brinkmann et al., 2004). Moreover, it causes lymphopenia in the blood via a rapid reduction in circulating lymphocytes (Rosen et al., 2003; Tiper et al., 2016). We see similar reduction in our treatments as well.

Given this circulating lymphopenia, it does not seem possible that circulating lymphocytes could populate the lungs and explain our findings. In addition CD90.1+Foxp3- cells in house dust mite sensitized animals show a CD69 hi, phenotype consistent with what is described for tissue resident memory cells (Figure 8—figure supplement 2). For this reason, we did not believe that the proposed parabiosis experiments were necessary.

4) Figure 6G, Figure 7—figure supplement 1: To validate the specificity of Derp1 tetramer staining, staining with a control tetramer (i.e., a tetramer loaded with an irrelevant peptide) should also be performed.

The specificity of the Derp1 tetramer has been extensively validated elsewhere (Hondowicz et al., 2015; Moon and Pepper, 2018). To corroborate this reagent ourselves we examined Derp1 tetramer staining in animals not previously exposed to house dust mite. These data support the specificity of this reagent (Figure 7—figure supplement 1).

5) Figure 8: It is not clear whether depletion of Thy1.1+ cells was properly assessed. It seems that anti-Thy1.1 (clone OX7) was used for flow cytometric analyses of T cells from anti-Thy1.1 (or isotype control) mAb injected animals. Because the anti-Thy1.1 mAb used for depletion should compete with the anti-Thy1.1 mAb used for flow cytometric analyses, the apparent absence of "Thy1.1+" cells in the treated group may not be due to their depletion but could simply result from the competition of the two clones of antibodies. This possibility should be ruled out.

To address this, we have used a distinct clone (Ox-7) to stain cells as compared to the antibody clone (19E12) which was used for depletion. We have included these data in Figure 9—figure supplement 1, which shows that the depletion antibody does not mask the staining epitope of Ox7 clone as well as another fluorescently tagged CD90.1 clone (HIS51).

In previous studies, many investigators, have taken advantage of the 10Bit mouse to deplete IL10 producing cells and demonstrated the loss of IL-10 protein by intracellular staining (Xin et al., 2018; Maynard et al., 2007). To further support the efficiency of depletion in our system, we have also included data that show intracellular IL-10 protein is reduced in depleted groups (Figure 9—figure supplement 1)

6) Figure 8C, D: The magnitude of the response of the control (yet HDM-immunized) group is much less than that of the HDM-immunized group shown in Figure 1B, C but rather similar to the non-immunized group. There must be something wrong.

Baseline responses to house dust mite (HDM) antigen can vary for several reasons. First, unlike peptide antigens or recombinant proteins HDM is purified from crushed insects. Different lots of commercial HDM can have different relative potency (RP). Lots were normalized based on Der p 1 protein, however endotoxin levels between lots varied. Second, the RP of a single lot can vary over its shelf life. For this reason, the baseline responses in different figures can vary. This point has been clarified in the text. Nonetheless, to address the variability between these experiments, we show data for two independent depletion experiments in Author response image 2.

**Author response image 2. respfig2:** 

Reviewer #3:The authors demonstrate the phenotype of lung resident Tr1-like cells during mite sensitization. They clearly show the stability and the origin of lung Tr1 cells using an in vitro assay system. However, some issues need to be further addressed to confirm and strengthen their claims on the role of Tr1 cells in the lung.Essential Revisions:1) The authors claim that the major source of lung IL-10 is Tr1-like cell after mite immunization because a half of IL10 positive cells is Foxp3-CD4^+^ (Figure 3B). However, total number of cells is also important to assess the contribution of IL-10 production. The total number of each cell population which produces IL-10 should be shown. The amount of IL-10 expression from each cell should also be addressed. In Figure 1, signal intensity of Thy1.1 (IL10) of Treg is higher than that of Tr1-like cells whereas the percentage is lower. As the original paper of the IL-10 reporter mice indicated that MFI of Thy1.1 can be used to estimate the IL-10 producing potential, they should summarize the MFI of Thy1.1 as the capacity of IL-10 expression, in addition to the percentage, of each cell population in order to support possible contribution of Tr1-like cells to the amount of lung IL-10.

To address this point, we have added assessments of absolute cell numbers and Thy1.1 MFI assessments to Figure 3.

2) They appear to have carried out the in vivo labeling experiment without immunization in Figure 2 because IL-10+ cells are not so high (Figure 2C). As the origin of Tr1 cells during inflammation is more important, they have to address the issue under immunization.

The image in question was indeed for a mouse that received PBS (a control mouse) and was provided to illustrate our gating strategy. To prevent any confusion, we have replaced this image with one from an immunized mouse (Figure 2).

3) In Figure 5, they compare IL-10 production with or without in vitro TCR stimulation. Is it possible to maintain primary T cells in vitro without stimulation for 5 days? They should show the percentage of living cells after culture. If unstimulated cells are not normal, it's difficult to conclude that IL-10 production is dependent on TCR stimulation.

The data in Figure 5 (in current manuscript it is Figure 6) are gated on live cells only. Nonetheless, the reviewer is correct that in the absence of TCR stimulation both the viable cell numbers as well as IL-10 levels are low. To accompany these data, we have added the viability data as Figure 6—figure supplement 1. We have also added statements to the text clarifying the inter-relatedness of these points. With regards to the second comment, the inability of CD90.1+ cells to produce IL-10 in vitro in the absence of antigenic stimulation has also been previously published in the original 10BiT paper (Maynard et al., 2007).

4) In figure 4G, they examine cytokine production from Thy1.1+ cells. Because IL-4 is a major Th2 cytokine, they should assess the production of IL-4.

To address this, we have included a Figure 7—figure supplement 2 which shows that the CD90.1+ cells are negative for IL-4, IL-13 and Gata3 via intracellular staining. We also show that memory Tr1 cells do not produce IL-4 in Figure 7I and therefore do not become Th2-like over time.

5) In Figure 7, they use FTY720 for inhibiting the egress of lymphocyte. As they only show the ratio of naïve/effector fraction in Figure 5F, it is difficult to understand the dynamics of CD4^+^ subsets. They should summarize the total number of each T cell subset with or without FTY720 treatment.

The total numbers of T cell subsets with or without FTY720 treatment has been added as Figure 8—figure supplement 1.

6) In Thy1.1 depletion experiments (Figure 8), not only Tr1 but also other IL-10 producing cell (e.g.; DCs, B cells and macrophages) population would be deleted. They should address the effects on other IL-10 producing cells, at least show the percentage and the number of other cell populations after anti-Thy1.1 treatment.

This information has been added as a Figure 9—figure supplement 2. However, when detecting intracellular IL-10, it is the CD4 T cells which are the greatest IL-10 producers and are most affected by the treatment with the depletion antibody.

[Editors’ note: the author responses to the re-review follow.]The manuscript has been improved but there are some remaining issues that need to be addressed before acceptance, as outlined below:The authors have properly responded to the reviewers' comments. One concern is their interpretation of the results in the depletion experiments in Figure 9 and Figure 10, which are related to whether IL-10-producing T cells indeed bear in vivo suppressive activity in the allergy model. This issue was raised by one of the reviewers, but the authors' response is not sufficiently convincing. In Figure 9A, B, there is still a possibility that depletion of Thy-1.1+Foxp3+ cells by anti-Thy-1.1 antibody is responsible for the exacerbation of the allergic response because these Foxp3+ cells are KLRG1+ (i.e., highly activated) even if the cell number is small as shown in Figure 4. Suppressive activity shown with in vitro induced Tr1 cells is not a proper support for their claim. However, the issue does not oppose their main claim that Tr-1-like cells do not possess suppressive activity on recall responses. Modification of their interpretation may be sufficient for final acceptance.

We agree with the editors and reviewers that our study does not exclude a functional role for CD90.1+Foxp3+ cells in suppressing inflammation. We have now made changes to the text which draw attention to this caveat in our results and in the discussion. Our changes convey our data more accurately. We have also updated the main manuscript file with these changes. The focus and main conclusion of our study remains on Tr1-like cells.